# Belief Propagation Neural Networks

**Jonathan Kuck**[1], **Shuvam Chakraborty**[1], **Hao Tang**[2], **Rachel Luo**[1], **Jiaming Song**[1],
**Ashish Sabharwal**[3], and **Stefano Ermon**[1]

[1]Stanford University  [2]Shanghai Jiao Tong University  [3]Allen Institute for Artificial Intelligence
{kuck,shuvamc,rsluo,tsong,ermon}@stanford.edu,
silent56@sjtu.edu.cn, ashishs@allenai.org

## Abstract

Learned neural solvers have successfully been used to solve combinatorial optimization and decision problems. More general counting variants of these problems, however, are still largely solved with hand-crafted solvers. To bridge this gap, we introduce belief propagation neural networks (BPNNs), a class of parameterized operators that operate on factor graphs and generalize Belief Propagation (BP). In its strictest form, a BPNN layer (BPNN-D) is a learned iterative operator that provably maintains many of the desirable properties of BP for any choice of the parameters. Empirically, we show that by training BPNN-D learns to perform the task better than the original BP: it converges 1.7x faster on Ising models while providing tighter bounds. On challenging model counting problems, BPNNs compute estimates 100's of times faster than state-of-the-art handcrafted methods, while returning an estimate of comparable quality.

## 1 Introduction

Probabilistic inference problems arise in many domains, from statistical physics to machine learning. There is little hope that efficient, exact solutions to these problems exist as they are at least as hard as NP-complete decision problems. Significant research has been devoted across the fields of machine learning, statistics, and statistical physics to develop variational and sampling based methods to approximate these challenging problems [13, 34, 48, 6, 38]. Variational methods such as Belief Propagation (BP) [31] are computationally efficient and have been particularly successful at providing principled approximations due to extensive theoretical analysis.

While BP provably lower bounds the partition function for classes of factor graphs, these bounds are not reliably tight. Handcrafting alternative algorithms that are specialized to problem domains and that provide bounds is laborious. We introduce belief propagation neural networks (BPNNs), a flexible neural architecture designed to estimate the partition function of a factor graph that leverage the theoretical analysis behind BP. BPNNs generalize BP and can thus provide more accurate estimates than BP when trained on a small number of factor graphs with known partition functions. During training BPNNs learn a modification to the standard BP message passing operations so that the final output is closer to the ground truth partition function. At the same time, BPNNs retain many of BP's properties, which results in more accurate estimates compared to general neural architectures. BPNNs are composed of iterative layers (BPNN-D) and an optional Bethe free energy layer (BPNN-B), both of which maintain the symmetries of BP under factor graph isomorphisms. BPNN-D is a parametrized iterative operator that strictly generalizes BP while preserving many of BP's guarantees. Like BP, BPNN-D is guaranteed to converge on tree structured factor graphs and return the exact partition function. For factor graphs with loops, BPNN-D computes a lower bound whenever the Bethe approximation obtained from fixed points of BP is a provable lower bound (with mild restrictions on BPNN-D). BPNN-B performs regression from the trajectory of beliefs (over a fixed number of iterations) to the partition function of the input factor graph. While this sacrifices

some guarantees, the additional flexibility introduced by BPNN-B generally improves estimation performance.

Experimentally, we show that on Ising models BPNN-D is able to converge faster than standard BP and frequently finds better fixed points that provide tighter lower bounds. BPNN-D generalizes well to Ising models sampled from a different distribution than seen during training and to models with nearly twice as many variables as seen during training, providing estimates of the log partition function that are significantly better than BP or a standard graph neural network (GNN) in these settings. We also perform experiments on community detection problems, where BP is known to perform well both empirically and theoretically, and show improvements over BP and a standard GNN. We then perform experiments on approximate model counting [46, 27, 28, 8], the problem of computing the number of solutions to a Boolean satisfiability (SAT) problem. Unlike the first two experiments it is very difficult for BP to converge in this setting. Still, we find that BPNN learns to estimate accurate model counts from a training set of 10's of problems and generalize to problems that are significantly harder for an exact model counter to solve. Compared to handcrafted approximate model counters, BP returns comparable estimates 100's times faster using GPU computation.

## 2 Background: Factor Graphs and Belief Propagation

In this section we provide background on factor graphs and belief propagation [31]. A factor graph is a representation of a discrete probability distribution that takes advantage of (conditional) independencies between variables to make the representation more compact. Belief propagation is a method for approximating the normalization constant, or partition function, of a factor graph. Let $p(\mathbf{x})$ be a discrete probability distribution defined over variables $\mathbf{x} = \{x_1, x_2, \ldots, x_n\}$ in terms of a factor graph as

$$p(\mathbf{x}) = \frac{1}{Z} \prod_{a=1}^{M} f_a(\mathbf{x}_a), \qquad Z = \sum_{\mathbf{x}} \left( \prod_{a=1}^{M} f_a(\mathbf{x}_a) \right). \qquad (1)$$

The factor graph is defined in terms of a set of factors $\mathbf{f} = \{f_1, f_2, \ldots, f_m\}$, where each factor $f_a$ takes a subset of variables $\mathbf{x_a} \subset \{x_1, x_2, \ldots, x_n\}$ as input and $f_a(\mathbf{x}_a) > 0$. $Z$ is the factor graph's normalization constant (or partition function). As a data structure, a factor graph is a bipartite graph with $n$ variables nodes and $M$ factor nodes. Factor nodes and variables nodes are connected if and only if the variable is in the scope of the factor.

**Belief Propagation**  Belief propagation performs iterative message passing between neighboring variable and factor nodes. Variable to factor messages, $m_{i \to a}^{(k)}(x_i)$, and factor to variable messages, $m_{a \to i}^{(k)}(x_i)$, are computed at every iteration $k$ as

$$m_{i \to a}^{(k)}(x_i) \coloneqq \prod_{c \in \mathcal{N}(i) \backslash a} m_{c \to i}^{(k-1)}(x_i), \text{ and } m_{a \to i}^{(k)}(x_i) \coloneqq \sum_{\mathbf{x}_a \backslash x_i} f_a(\mathbf{x}_a) \prod_{j \in \mathcal{N}(a) \backslash i} m_{j \to a}^{(k)}(x_j). \qquad (2)$$

These messages are vectors over the states of variable $x_i$. The BP algorithm estimates approximate marginal probabilities over the sets of variables $\mathbf{x}_a$ associated with each factor $f_a$. We denote the belief over variables $\mathbf{x}_a$, after message passing iteration $k$ is complete, as $b_a^{(k)}(\mathbf{x}_a) = \frac{f_a(\mathbf{x}_a)}{z_a} \prod_{i \in \mathcal{N}(a)} m_{i \to a}^{(k)}(x_i)$ with $z_a = \sum_{\mathbf{x}_a} f_a(\mathbf{x}_a) \prod_{i \in \mathcal{N}(a)} m_{i \to a}^{(k)}(x_i)$. Similarly, BP computes beliefs at each variable as $b_i^{(k)}(x_i) = \frac{1}{z_i} \prod_{a \in \mathcal{N}(i)} m_{a \to i}^{(k)}(x_i)$. The belief propagation algorithm proceeds by iteratively updating variable to factor messages and factor to variable messages until they converge to fixed values, referred to as a fixed point of Equations 2, or a predefined maximum number of iterations is reached. At this point the beliefs are used to compute a variational approximation of the factor graph's partition function. This approximation, originally developed in statistical physics, is known as the Bethe free energy $F_{\text{Bethe}} = U_{\text{Bethe}} - H_{\text{Bethe}} \approx -\ln Z$ [10]. It is defined in terms of the Bethe average energy $U_{\text{Bethe}} \coloneqq -\sum_{a=1}^{M} \sum_{\mathbf{x}_a} b_a(\mathbf{x}_a) \ln f_a(\mathbf{x}_a)$ and the Bethe entropy $H_{\text{Bethe}} \coloneqq -\sum_{a=1}^{M} \sum_{\mathbf{x}_a} b_a(\mathbf{x}_a) \ln b_a(\mathbf{x}_a) + \sum_{i=1}^{N} (d_i - 1) \sum_{x_i} b_i(x_i) \ln b_i(x_i)$, where $d_i$ is the degree of variable node $i$.

## 3 Belief Propagation Neural Networks

We design belief propagation neural networks (BPNNs) as a family of graph neural networks that subsume BP. Like BP, BPNNs take a factor graph as input and output an estimate of the factor graph's log partition function. Unlike standard graph neural networks (GNNs), BPNNs do not resend

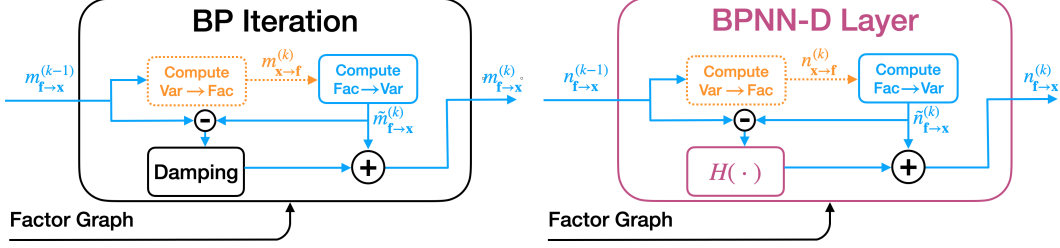

Figure 1: Left: one iteration of BP with damping. Right: one iteration of BPNN-D with learnable $H(\cdot)$.

messages between nodes, a property taken from BP known as avoiding 'double counting' the evidence. This property guarantees that BPNN-D described below is exact on trees (Theorem 3). BPNN-D is a strict generalization of BP (Proposition 1), but is still guaranteed to give a lower bound to the partition function upon convergence for a class of factor graphs (Theorem 3) by finding fixed points of BP (Theorem 2). Like BP, BPNN preserves the symmetries inherent to factor graphs (Theorem 4).

BPNNs consist of two parts. First, iterative BPNN layers output messages, analogous to standard BP. These messages are used to compute beliefs using the same equations as for BP. Second, the beliefs are passed into a Bethe free energy layer (BPNN-B) which generalizes the Bethe approximation by performing regression from beliefs to $\ln(Z)$. Alternatively, when the standard Bethe approximation is used in place of BPNN-B, BPNN provides many of BP's guarantees.

**BPNN Iterative Layers**   BPNN iterative layers are flexible neural operators that can operate on beliefs or message in a variety of ways. Here, we focus on a specific variant, BPNN-D, due to its strong convergence properties, and we refer the reader to Appendix C for information on other variants. The BPNN iterative damping layer (BPNN-D, shown in Fig 1) modifies factor-to-variable messages (Equation 2) using the output of a learned operator $H : \mathbb{R}^{\sum_{i=1}^{n} d_i |X_i|} \to \mathbb{R}^{\sum_{i=1}^{n} d_i |X_i|}$, where $d_i$ denotes the degree and $|X_i|$ the cardinality of variable $X_i$. This learned operator $H(\cdot)$ takes as input the difference between iterations $k-1$ and $k$ of every factor-to-variable message, and modifies these differences jointly. It can thus be much richer than a scalar multiplier. BPNN-D factor-to-variable messages are given by

$$n_{a\to i}^{(k)} = \tilde{n}_{a\to i}^{(k)} + \Delta_{a\to i}^{(k)}, \text{ and } \tilde{n}_{a\to i}^{(k)}(x_i) \coloneqq \sum_{\mathbf{x}_a \setminus x_i} f_a(\mathbf{x}_a) \prod_{j\in\mathcal{N}(a)\setminus i} m_{j\to a}^{(k)}(x_j). \quad (3)$$

Let $\Delta_{\mathbf{f}\to\mathbf{x}}^{(k)} = H\big(n_{\mathbf{f}\to\mathbf{x}}^{(k-1)} - \tilde{n}_{\mathbf{f}\to\mathbf{x}}^{(k)}\big)$ denote the result of applying $H(\cdot)$ to all factor-to-variable message differences.[1] Then $\Delta_{a\to i}^{(k)}$ is the output corresponding to the modified $a \to i$ message difference.

Variable-to-factor messages are unchanged from Eq. 18, except for taking messages $n_{a\to i}^{(k)}$ as input,

$$n_{i\to a}^{(k)}(x_i) \coloneqq \prod_{c\in\mathcal{N}(i)\setminus a} n_{c\to i}^{(k-1)}(x_i). \quad (4)$$

Damping is a standard technique for improving the convergence of BP when updates tend to 'overshoot' (see Section B.1 in the Appendix). Standard damped BP is recovered if $H$ is an elementwise function $H(x) = \alpha x$. Thus:

**Proposition 1.** *BPNN-Ds subsume BP and damped BP as a strict generalization.*

For non-trivial choices of $H(\cdot)$, whether BPNN preserves the fixed points of BP or introduces any new ones turns out to depend only on the set of fixed points of $H(\cdot)$ itself, i.e., $\{x \mid H(x) = x\}$. As we show next, this property allows us to easily enforce that every fixed point of BP is also a fixed point of BPNN-D (Theorem 1), or vice versa (Theorem 2).[2]

**Theorem 1.** *If zero is a fixed point of $H(\cdot)$, then every fixed point of BP is also a fixed point of BPNN-D.*

**Theorem 2.** *If $H(\cdot)$ does not have any non-zero fixed points, then every fixed point of BPNN-D is also a fixed point of BP.*

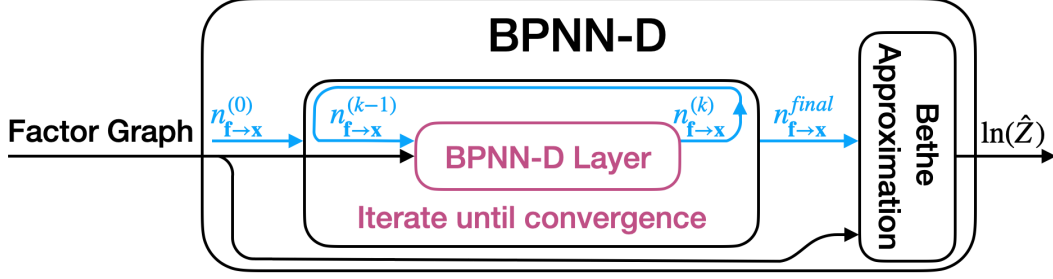

Figure 2: Computation graph of BPNN-D applied iteratively and followed by the Bethe approximation.

Combining Theorems 1 and 2, we obtain Corollary 2.1.

**Corollary 2.1.** *If zero is the unique fixed point of $H(\cdot)$, then the fixed points of BP and BPNN-D are identical. This property is satisfied when $H(x) = x + \bar{H}(x) - \bar{H}(\mathbf{0})$ for any invertible function $\bar{H}(\cdot)$.*

Note that a broad class of highly expressive learnable operators are invertible [7]. Enforcing that every fixed point of BPNN-D is also a fixed point of BP is particularly useful, as it immediately follows that BPNN-D returns a lower bound whenever the Bethe approximation obtained from fixed points of BP returns a provable lower bound (Theorem 3). When a BPNN-D layer is applied iteratively until convergence, fast convergence is guaranteed for tree structured factor graphs (Proposition 2).

**Theorem 3.** *If zero is the unique fixed point of $H(\cdot)$, the Bethe approximation computed from beliefs at a fixed point of BPNN-D (1) is exact for tree structured graphs and (2) lower bounds the partition function of any factor graph with binary variables and log-supermodular potential functions.*

**Proposition 2.** *BPNN-D converges within $\ell$ iterations on tree structured factor graphs with height $\ell$.*

As mentioned, BPNN iterative layers are flexible and can additionally be modified to operate directly on message values or factor beliefs at the expense of guarantees (see Appendix C).

**Bethe Free Energy Layer (BPNN-B).** When convergence to a fixed point is unnecessary, we can increase the flexibility of our architecture by building a K-layer BPNN from iterative layers that do not share weights. Additionally we define a Bethe free energy layer (BPNN-B, Equation 5) using two MLPs that take the trajectories of beliefs (across all $K$ layers) from each factor and variable as input and output scalars:

$$f_{\text{BPNN}}(G_{\text{factor}}) = \sum_{i=1}^{n} \text{MLP}_{BV} \left[ \underset{k=1}{\overset{K}{\text{CONCAT}}} \left( (d_i - 1)b_i^{(k)}(x_i) \ln b_i^{(k)}(x_i) \right) \right] +$$

$$\frac{1}{|\mathbf{x}_a|!} \sum_{a=1}^{M} \sum_{\sigma \in S_{|\mathbf{x}_a|}} \text{MLP}_{BF} \left[ \underset{k=1}{\overset{K}{\text{CONCAT}}} \left( \sigma \left( b_a^{(k)}(\mathbf{x}_a) \ln f_a(\mathbf{x}_a) \right), \sigma \left( -b_a^{(k)}(\mathbf{x}_a) \ln b_a^{(k)}(\mathbf{x}_a) \right) \right) \right]. \quad (5)$$

This parameterization subsumes the standard Bethe approximation, so we can initialize the parameters of $f_{\text{BPNN}}$ to output the Bethe approximation computed from the final layer beliefs (see the appendix for details). Note that $|\mathbf{x}_a|$ is the number of variables in the scope of factor $a$, $S_{|\mathbf{x}_a|}$ denotes the symmetric group (all permutations of $\{1, 2, \ldots, |\mathbf{x}_a|\}$), and the permutation $\sigma$ is applied to the dimensions of all $2k$ concatenated terms. We ensure that BPNN preserves the symmetries of BP (Theorem 4) by passing all factor permutations through $\text{MLP}_{BF}$ and averaging the result.

**BPNN Preserves the Symmetries of BP.** BPNN is designed so that equivalent input factor graphs are mapped to equivalent outputs. This is a property that BP satisfies by default. Standard GNNs are also designed to satisfy this property, however the notion of 'equivalence' between graphs is different than 'equivalence' between factor graphs. In this section we formalize these statements.

Graph isomorphism defines an equivalence relationship between graphs that is respected by standard GNNs. Two isomorphic graphs are structurally equivalent and indistinguishable if the nodes are appropriately matched. More formally, there exists a bijection between nodes (or their indices) in the two graphs that defines this matching. Standard GNNs are designed so that output node representations are equivariant to the input node indexing; the indexing of output node representations matches the indexing of input nodes. Output node representations of a GNN run on two isomorphic

graphs can be matched using the same bijection that defines the isomorphism. Further, standard GNNs are designed to map isomorphic graphs to the same graph-level output representation. These two properties are achieved by using a message aggregation function and a graph-level output function that are both invariant to node indexing.

We formally define factor graph isomorphism in Definition 1 (Appendix A). This equivalence relationship is more complicated than for standard graphs because factor potentials define a structured relationship between factor and variable nodes. As in a standard graph, variable nodes are indexed globally $(X_1, X_2, \ldots, X_n)$ in the representation of a factor graph. Additionally, variable nodes are also indexed locally by factors that contain them. This is required because each factor dimension (note that factors are tensors) corresponds to a unique variable, unless the factor happens to be symmetric. Local variable indices define a mapping between factor dimensions and the variables' global indices. These local variable indices lead to additional bijections in the definition of isomorphic factor graphs (condition 2 in Definition 1). Note that standard GNNs do not respect factor graph isomorphisms because of these additional bijections.

In contrast to standard GNNs, BP respects factor graph isomorphisms. When BP is run on two isomorphic factor graphs for the same number of iterations with constant message initialization[3] the output beliefs and messages satisfy bijections corresponding to those of the input factor graphs. Specifically, messages are equivariant to global node indexing (Lemma 1), variable beliefs are equivariant to global variable node indexing (Lemma 2), and factor beliefs are equivariant to global factor node indexing and local variable node indexing within factors (Lemma 3). We refer to the above properties as *equivariances of BP* under factor graph isomorphisms. We show that these properties also apply to BPNN-D when $H(\cdot)$ is equivariant to global node indexing. The Bethe approximation obtained from isomorphic factor graphs is identical, when BP is run for the same number of iterations with constant message initialization[3]. BPNN-B also satisfies this property because it is, by design, invariant to local variable indexing within factors (Lemma 4). Together, these properties lead to the following:

**Theorem 4.** *If $H(\cdot)$ is equivariant to global node indexing, then (1) BPNN-D messages and beliefs preserve the equivariances of BP under factor graph isomorphisms and (2) BPNN-B is invariant under factor graph isomorphisms.*

## 4 Experiments

We trained BPNN to estimate the partition function of factor graphs from a variety of domains. First, experiments on synthetic Ising models show that BPNN-D can learn to find better fixed points than BP and converge faster. Additionally, BPNN generalizes to Ising models with nearly twice as many variables as those seen during training and that were sampled from a different distribution. Second, experiments and an ablation study on the stochastic block model from community detection show that maintaining properties of BP in BPNN improves results over standard GNNs. Finally, model counting experiments performed on real world SAT problems show that BPNN can learn from 10's of training problems, generalize to problems that are harder for an exact model counter, and compute estimates 100's of times faster than handcrafted approximate model counters. We implemented our BPNN and the baseline GNN using PyTorch Geometric [19]. See Appendix B.2 for details on the GNN. In all our experiments, we initialized the BPNN to output the Bethe approximation obtained by running BP for a specified number of iterations. We used the mean squared error between the BPNN prediction and the ground truth log partition function as our training loss.

### 4.1 Ising Models

We followed a common experimental setup used to evaluate approximate integration methods [21, 17]. We randomly generated grid structured attractive Ising models whose partition functions can be computed exactly using the junction tree algorithm [33] for training and validation. BP computes a provable lower bound for these Ising models [41]. This family of Ising models is only slightly more general than the one studied in [30], where BP was proven to quickly converge to the Bethe free energy's global optimum. We found that an iterative BPNN-D layer was able to converge faster than standard BP and could find tighter lower bounds for these problems. Additionally we trained a 10 layer BPNN and evaluated its performance against a 10 layer GNN architecture (details in Appendix).

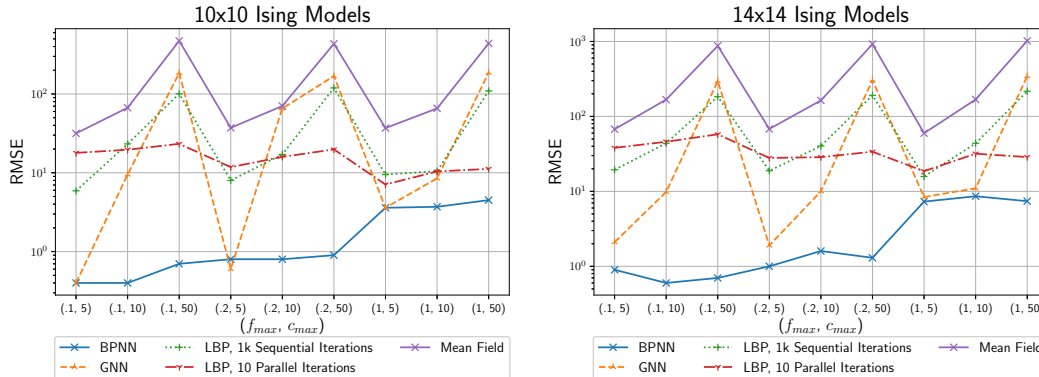

Figure 3: Each point represents the root mean squared error (RMSE, y-axis) of the specified method on a test set of 50 Ising models sampled with the parameters $f_{\max}$ and $c_{\max}$ (x-axis). The leftmost point shows results for test data drawn from the same distribution as training. BPNN significantly improves upon loopy belief propagation (LBP) for both in and out of distribution data. BPNN also significantly outperforms GNN on out of distribution data and larger models.

Compared to the GNN, BPNN has improved generalization when tested on larger Ising models and Ising models sampled from a different distribution than seen during training.

**Improved Lower Bounds and Faster Convergence.** We trained an iterative BPNN-D layer to lower bound the partition function on a training set of 50 random Ising models of size 10x10 (100 variables). (See the appendix for further details.) We then ran the learned BPNN-D and standard BP on a validation set of 50 Ising models. We empirically verified that BPNN-D found fixed points corresponding to tighter lower bounds than BP, and that it found them faster than standard BP. BPNN-D converged on all 50 models, while BP failed to converge within 200 iterations for 6 of the models. We recorded the number of iterations that BPNN-D and BP run with parallel updates took to converge, defined as a maximum factor-to-variable message difference of $10^{-5}$. BPNN-D had a median improvement ratio of 1.7x over BP, please refer to the appendix for complete convergence plots. Among the 44 models where BP converged, the RMSE between the exact log partition function and BPNN-D's estimate was .97 compared with 7.20 for BP. For 10 of the 44 models, BPNN-D found fixed points corresponding to lower bounds on the log partition function that were larger (i.e., better) than BP's by 3 to 22 (corresponding to bounds on the partition function that were 20 to $e^{22}$ times larger). In contrast, the log lower bound found by BP was never larger than the bound found by BPNN-D by more than 1.7.

**Out of Distribution Generalization.** We tested BPNN's ability to generalize to larger factor graphs and to shifts in the test distribution. Again we used a training set of 50 Ising models of size 10x10 (100 variables). We sampled test Ising models from distributions with generative parameters increased by factors of 2 and 10 from their training values (see appendix for details) and with their size increase to 14x14 (for 196 variables instead of the 100 seen during training). For this experiment we used a BPNN architecture with 10 iterative layers whose weights were not tied and with MLPs that operate on factor messages (without a BPNN-B layer). As a baseline we trained a 10 layer GNN (maximally powerful GIN architecture) with width 4 on the same dataset. We also compute the Bethe approximation from running standard loopy belief propagation and the mean field approximation. We used the libDAI [35] implementation for both. We tested loopy belief propagation with and without damping and with both parallel and sequential message update strategies. We show results for two settings whose estimates of the partition function differ most drastically: (1) run for a maximum of 10 iterations with parallel updates and damping set to .5, and (2) run for a maximum of 1000 iterations with sequential updates using a random sequence and no damping. Full test results are shown in Figure 3. The leftmost point in the left figure shows results for test data that was drawn from the same distribution used for training the BPNN and GNN. The BPNN and GNN perform similarly for data drawn from the same distribution seen during training. However, our BPNN significantly outperforms the GNN when the test distribution differs from the training distribution and when generalizing to the larger models. Our BPNN also significantly outperforms loopy belief propagation, both for test data drawn from the training distribution and for out of distribution data.

| Stochastic Block Model RMSE | | | | |
|---|---|---|---|---|
| BP | GNN | BPNN-DC | BPNN-NI | BPNN |
| Train/Val | Train/Val | Train/Val | Train/Val | Train/Val |
| 12.55/11.14 | 7.33/7.93 | 7.04/8.43 | 4.43/5.63 | **4.16/4.15** |

Table 1: RMSE of SBM $\ln(Z)$ estimates. BPNN outperforms BP, GNN, and ablated versions of BPNN.

## 4.2 Stochastic Block Model

The Stochastic Block Model (SBM) is a generative model describing the formation of communities and is often used to benchmark community detection algorithms [1]. While BP does not lower bound the partition functions of associated factor graphs for SBMs, it has been shown that BP asymptotically (in the number of nodes) reaches the information theoretic threshold for community recovery on SBMs with fewer than 4 communities [1]. We trained a BPNN to estimate the partition function of the associated factor graph and observed improvements over estimates obtained by BP or a maximally powerful GNN, which lead to more accurate marginals that can be used to better quantify uncertainty in SBM community membership. We refer the reader to Appendix F for a formal definition of SBMs as well as our procedure for constructing factor graphs from a sampled SBM.

**Dataset and Methods** In our experiments, we consider SBMs with 2 classes and 15-20 nodes, so that exact inference is possible using the Junction Tree algorithm. In this non-asymptotic setting, BP is a strong baseline and can almost perfectly recover communities [14], but is not optimal and thus does not compute exact marginals or partition functions. For training, we sample 10 two class SBMs with 15 nodes, class probabilities of .75 and .25, and edge probability of .93 within and .067 between classes along with four such graphs for validation. For each graph, we fix each node to each class and calculate the exact log partition using the Junction Tree Algorithm, producing 300 training and 120 validation graphs. We explain in Appendix F how these graphs can be used to calculate marginals.

To estimate SBM partition functions, we trained a BPNN with 30 iterative BPNN layers that operate on messages (see Appendix C), followed by a BPNN-B layer. Since BP does not provide a lower bound for SBM partitions, we took advantage of BPNN's flexibility and chose greater expressive power over BPNN-D's superior convergence properties. We compared against BP and a GNN as baseline methods. Additionally, we performed 2 ablation experiments. We trained a BPNN with a BPNN-B layer that was not permutation invariant to local variable indexing, by removing the sum over permutations in $S_{|\mathbf{x}_a|}$ from Equation 5 and only passing in the original beliefs. We refer to this non-invariant version as BPNN-NI. We then forced BPNN-NI to 'double count' messages by changing the sums in Equations 22 and 23 to be over $j \in \mathcal{N}(a)$. We refer to this non-invariant version that performs double counting as BPNN-DC. We refer the reader to Appendix F for further details on models and training.

**Results** As shown in Table 1, BPNN provides the best estimates for the partition function. Critically, we see that not 'double counting' messages and preserving the symmetries of BP are key improvements of BPNN over GNN. Additionally, BPNN outperforms BP and GNN on out of distribution data and larger graphs and can learn more accurate marginals. We refer the reader to Appendix F for more details on these additional experiments.

## 4.3 Model Counting

In this section we use a BPNN to estimate the number of satisfying solutions to a Boolean formula, a challenging problem for BP which generally fails to converge due to the complex logical constraints and 0 probability states. Computing the exact number of satisfying solutions (exact model counting) is a #P-complete problem [47]. Model counting is a fundamental problem that arises in many domains including probabilistic reasoning [40, 9], network reliability [16], and detecting private information leakage from programs [11]. However, the computational complexity of exact model counting has led to a significant body of work on approximate model counting [46, 27, 28, 8, 20, 18, 24, 3, 5, 44], with the goal of estimating the number of satisfying solutions at a lower computational cost.

**Training Setup.** All BPNNs trained in this section were composed of 5 BPNN-D layers followed by a BPNN-B layer and were trained to predict the natural logarithm of the number of satisfying solutions to an input formula in CNF form. This is accomplished by converting the CNF formula into a factor graph whose partition function is the number of satisfying solutions to the input formula. We evaluated the performance of our BPNN using benchmarks from [44], with ground truth model counts obtained using DSharp [37]. The benchmarks fall into 7 categories, including network QMR problems (Quick Medical Reference) [26], network grid problems, and bit-blasted versions of satisfiability

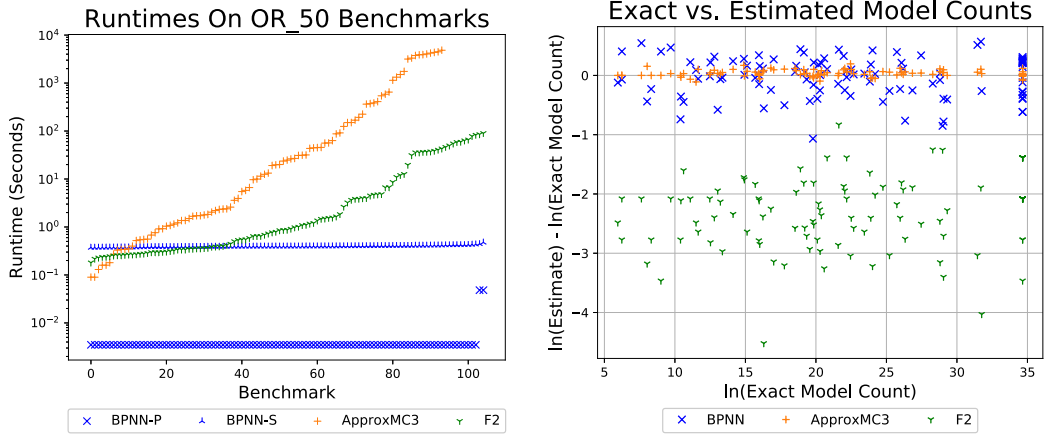

Figure 4: Left: cactus plot of runtimes for the 105 instances in the 'or_50' category solved by BPNN, F2, and ApproxMC3. BPNN-P denotes the time taken to run BPNN in parallel on a GPU divided by the number of instances per batch (batch size=103). Median speedups of BPNN-P over F2 and ApproxMC among the plotted benchmarks are 248 and 3,689 respectively. BPNN-S denotes the time taken to run BPNN sequentially on each instance (using a CPU). Median speedups of BPNN-S over F2 and ApproxMC among the plotted benchmarks are 2.2 and 32, resp. While BPNN solved each instance within 1 second, ApproxMC3 *timed out* on 12 instances (out of 105) after 5000 seconds, which are not plotted. Right: error in estimated log model count (base e) plotted against the exact model count for 'or_50' training and validation benchmarks. BPNN's validation RMSE was .30 on this category compared with a RMSE of 2.5 for F2.

modulo theories library (SMTLIB) benchmarks [12]. Each category contains 14 to 105 problems allocated for training and validation. See the appendix for additional details on training, the dataset, and our use of minimal independent support variable sets.

**Baseline Approximate Model Counters.** For comparison we ran two state-of-the-art approximate model counters on all benchmarks, ApproxMC3 [12, 44] and F2 [4, 5]. ApproxMC3 is a randomized hashing algorithm that returns an estimate of the model count that is guaranteed to be within a multiplicative factor of the exact model count with high probability. F2 gives up the probabilistic guarantee that the returned estimate will be within a multiplicative factor of the true model count in return for significantly increased computational efficiency. We also attempted to train a GNN, using the architecture from [43] adapted from classification to regression. We used the author's code, slightly modified to perform regression, but were not successful in achieving non-trivial learning.

**BPNNs Provide Excellent Computational Efficiency.** Figure 4 shows runtimes and estimates for BPNN, ApproxMC3, and F2 on all benchmarks from the category 'or_50'. BPNN is signficantly faster than both F2 and ApproxMC. BPNN provides median speedups of 2.2 and 32 over F2 and ApproxMC3 when all methods are run using a CPU. When BPNN is allowed to run in parallel on a GPU, it provides median speedups of 248 and 3,689 over F2 and and ApproxMC3. Additionally, BPNN's estimates are significantly tighter than F2's, with a RMSE for BPNN of .30 compared with 2.5 for F2. Please see the appendix for further runtime comparisons between methods.

**Learning from Limited Data.** We trained a separate BPNN on a random sampling of 70% of the problems in each training category. This gave each BPNN only 9 to 73 benchmarks to learn from. In contrast, prior work has performed approximate model counting on Boolean formulas in disjunctive normal form (DNF) by creating a large training set of 100k examples whose model counts can be approximated with an efficient polynomial time algorithm [2]. Such an algorithm does not exist for model counting on CNF formulas, making this approach intractable. Nonetheless, BPNN achieves training and validation RMSE comparable to or better than F2 across the range of benchmark categories (see the appendix for complete results). This demonstrates that BPNNs can capture the distribution of diverse families of SAT problems in an extremely data limited regime.

**Generalizing from Easy Data to Hard Data.** We repeated the same experiment from the previous paragraph, but trained each BPNN on the 70% of the problems from each category that DSharp solved fastest. Validation was performed on the remaining 30% of problems that took longest for DSharp to solve. These hard validation sets are significantly more challenging for Dsharp. The median runtime in each category's hard validation set is 4 to 15 times longer than the longest runtime in each corresponding easy training set. Validation RMSE on these hard problems was within 33%

of validation error when trained and validated on a random sampling for 3 of the 7 categories. This demonstrates that BPNNs have the potential to be trained on available data and then generalize to related problems that are too difficult for any current methods. See the appendix for complete results.

**Learning Across Diverse Domains.** We trained a BPNN on a random sampling of 70% of problems from all categories, spanning network grid problems, bit-blasted versions of SMTLIB benchmarks, and network DQMR problems. The BPNN achieved a final training RMSE of 3.9 and validation RMSE of 5.3, demonstrating that the BPNN is capable of capturing a broad distribution that spans multiple domains from a small training set.

## 5    Related Work

[2] use a graph neural network to perform approximate weighted disjunctive normal form (DNF) counting. Weighted DNF counting is a #P-complete problem. However, in contrast to model counting on CNF formulas, there exists an $O(nm)$ polynomial time approximation algorithm for weighted DNF counting (where $n$ is the number of variables and $m$ is the number of clauses). The authors leverage this to generate a large training dataset of 100k DNF formulas with approximate solutions. In comparison, our BPNN can learn and generalize from a very small training dataset of less than 50 problems. This result provides the significant future work alluded to in the conclusion of [2].

Recently, [42] designed a graph neural network that operates on factor graphs and exchanges messages with BP to perform error correction decoding. In contrast, BPNN-D preserves all of BP's fixed point, computes the exact partition function on tree structured factor graphs, and returns a lower bound whenever the Bethe approximation obtained from fixed points of BP is a provable lower bound. All BPNN layers preserve BP's symmetries (invariances and equivariances) to permutations of both variable and factor indices. Finally BPNN avoids 'double counting' during message passing.

Prior work has shown that neural networks can learn how to solve NP-complete decision problems and optimization problems [43, 39, 23]. [53] perform marginal inference in relatively small graphical models using GNNs. [22] consider improving message passing in expectation propagation for probabilistic programming, when users can specify arbitrary code to define factors and the optimal updates are intractable. [50] consider learning Markov random fields and address the problem of estimating marginal likelihoods (generally intractable to compute precisely). They use a transformer network that is faster than LBP but computes comparable estimates. This allows for faster amortized inference during training when likelihoods must be computed at every training step. In contrast, BPNNs significantly outperform LBP and generalize to out of distribution data.

## 6    Conclusion

We introduced belief propagation neural networks, a strict generalization of BP that learns to find better fixed points faster. The BPNN architecture resembles that of a standard GNN, but preserves BP's invariances and equivariances to permutations of variable and factor indices. We empirically demonstrated that BPNNs can learn from tiny data sets containing only 10s of training points and generalize to test data drawn from a different distribution than seen during training. BPNNs significantly outperform loopy belief propagation and standard graph neural networks in terms of accuracy. BPNNs provide excellent computational efficiency, running orders of magnitudes faster than state-of-the-art randomized hashing algorithms while maintaining comparable accuracy.

## Broader impact

This work makes both a theoretical contribution and a practical one by advancing the state-of-the-art in approximate inference on some benchmark problems. Our theoretical analysis of neural fixed point iterators is unlikely to have a direct impact on society. BPNN, on the other hand, can make approximate inference more scalable. Because approximate inference is a key computational problem underlying, for example, much of Bayesian statistics, it is applicable to many domains, both beneficial and harmful to society. Among the beneficial ones, we have applications of probabilistic inference to medical diagnosis and applications of model counting to reliability, safety, and privacy analysis.

## Acknowledgements

We thank Tri Dao, Ines Chami, and Shengjia Zhao for helpful discussions and feedback. Research supported by NSF (#1651565, #1522054, #1733686), ONR (N00014-19-1-2145), AFOSR (FA9550-19-1-0024), and FLI.

## Footnotes

[1] We use message subscripts $\mathbf{f}$ and $\mathbf{x}$ to denote sets of messages, e.g., $\overline{n}_{\mathbf{f}\to\mathbf{x}}^{(k)} = \{\overline{n}_{a\to i}^{(k)} : f_a \in \mathbf{f}, x_i \in \mathbf{x}\}$.

[2] For lack of space, all proofs are deferred to Appendix A.

[3]Any message initialization can be used, as long as initial messages are equivariant, see Lemma 1.

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
