[Supplementary Material]

# A PROOFS

*Theorem 1.* Every fixed point of BP satisfies $\tilde{m}_{a\to i}^{(k)} = \overline{m}_{a\to i}^{(k-1)}$ by definition. The computation of $\tilde{n}_{a\to i}^{(k)}$ from $\overline{n}_{a\to i}^{(k-1)}$ (Equations 22 and 23) is identical to the computation of $\tilde{m}_{a\to i}^{(k)}$ from $\overline{m}_{a\to i}^{(k-1)}$ in standard BP (Equations 18 and 19). Therefore, every fixed point of BP satisfies $\overline{n}_{a\to i}^{(k-1)} = \tilde{n}_{a\to i}^{(k)}$ and is also a fixed point of BPNN-D when $H(0) = 0$ $\qquad\square$

*Theorem 2.* Every fixed point of BPNN-D satisfies $\overline{n}_{a\to i}^{(k)} = \overline{n}_{a\to i}^{(k-1)}$ by definition. Equation 22 gives $\overline{n}_{a\to i}^{(k-1)} - \tilde{n}_{a\to i}^{(k)} = \Delta_{a\to i}^{(k)} = H\big(\overline{n}^{(k-1)} - \tilde{n}^{(k)}\big)_{a\to i}$. Given the restriction on $H(\cdot)$ that $H(x) = x$ only if $x = 0$, it follows that $\overline{n}_{a\to i}^{(k-1)} - \tilde{n}_{a\to i}^{(k)} = 0$. This is a fixed point of BP by definition as the computation of $\tilde{n}_{a\to i}^{(k)}$ from $\overline{n}_{a\to i}^{(k-1)}$ is identical to the computation of $\tilde{m}_{a\to i}^{(k)}$ from $\overline{m}_{a\to i}^{(k-1)}$ in standard BP (Equations 18 and 19). $\qquad\square$

*Theorem 3.* If zero is the unique fixed point of $H(\cdot)$, then the fixed points of BPNN-D and BP are identical by Theorems 1 and 2. Therefore, (1) the Bethe approximation obtained from fixed points of BPNN-D on tree structured factor graphs is exact because it is exact for fixed points of BP [31] (or see [36][p.27] for a detailed proof). (2) Ruozzi [41][p.8] prove in Corollary 4.2 that the Bethe approximation at any fixed point of BP is a lower bound on the partition function for factor graphs with binary variables and log-supermodular potential functions. so it follows that the Bethe approximation at any fixed point of BPNN-D lower bounds the partition function. $\qquad\square$

*Proposition 2.* If we consider a BPNN with weight tying, then regardless of the number of iterations or layers, the output messages are the same if the input messages are the same. Without loss of generality, let us first consider any node $r$ as the root node, and consider all the messages on the path from the leaf nodes through $r$. Let $d_{r,i}$ denote the depth of the sub-tree with root $i$ when we consider $r$ as the root (e.g. for a leaf node $i$, $d_{r,i} = 1$). We use the following induction argument:

- At iteration 1, the message from all nodes with $d_{r,i} = 1$ to their parents will be fixed for subsequent iterations since the inputs to the BPNN for these messages are the same.

- If at iteration $t - 1$, the message from all nodes with $d_{r,i} \leq t - 1$ to their parents are fixed for all subsequent iterations, then the inputs to the BPNN for all the messages from all nodes with $d_{r,i} = t$ to their parents will be fixed (since they depend on lower level messages that are fixed). Therefore, at iteration $t$, the messages from all the nodes with $d_{r,i} \leq t$ to their parents will be fixed because of weight tying between BPNN layers.

- The maximum tree depth is $l$, so $\max_i d_{r,i} \leq l$. From the induction argument above, after at most $l$ iterations, all the messages along the path from leaf nodes to $r$ will be fixed.

Since the BPNN layer performs the operation over all nodes, this above argument is valid for all nodes when we consider them as root nodes. Therefore, all messages will be fixed after at most $l$ iterations, which completes the proof. $\qquad\square$

**Isomorphic Factor Graphs** To prove Theorem 4 we define isomorphic factor graphs, an equivalence relation among factor graph representations, and break Theorem 4 into the lemmas in this section. Standard GNNs are built on the assumption that isomorphic graphs should be mapped to the same representation and non-isomorphic graphs should be mapped to different representations [51]. This is a challenging goal, in fact [51][p.4] prove in Lemma 2 that any GNN that aggregates messages from 1-hop neighbors is, at most, as discriminative as the Weisfeiler-Lehman (WL) graph isomorphism test. Xu et al. [51] go on to propose a provably 'maximally powerful' GNN, one that maps isomorphic graphs to the same representation and maps non-isomorphic graphs to different representations whenever the WL test maps them to different representations, which is the best result possible for this class of graph neural networks that aggregate messages from 1-hop neighbors. The input to a standard GNN is a graph represented by an adjacency matrix whose i-th row and column correspond to the i-th node. Nodes and edges may have corresponding features. The GNN in [51] was designed to map isomorphic graphs to the same representation by outputting learned node

representations that are equivariant to the input node indexing and a graph wide representation that is invariant to input node indexing.

The input to a BPNN is a factor graph. With the same motivation as for standard GNNs, BPNNs should map isomorphic factor graphs to the same output representation. A factor graph is represented as[4] $G = (A, F^p, F^{idx})$. $A \in \{0,1\}^{M \times N}$ is an adjacency matrix over $M$ factor nodes and $N$ variable nodes, where[5] $A_{ai} = 1$ if the i-th variable is in the scope of the a-th factor and $A_{ai} = 0$ otherwise. $F^p$ is an ordered list of $M$ factor potentials, where the a-th factor potential, $F_a^p$, corresponds to the a-th factor (row) in $A$ and is represented as a tensor with one dimension for every variable in the scope of $F_a^p$. $F^{idx}$ is an ordered list of ordered lists that locally indexes variables within each factor. $F_a^{idx}$ is an ordered list specifying the local indexing of variables within the a-th factor (in $A$ and $F^p$). $F_{ak}^{idx} = i$ specifies that the k-th dimension of the tensor $F_a^p$ corresponds to the i-th variable (column) in $A$. We define two factor graphs to be isomorphic when they meet the conditions of Definition 1.

**Definition 1.** *Factor graphs $G = (G(A), G(F^p), G(F^{idx}))$ and $G' = (G'(A), G'(F^p), G'(F^{idx}))$ with $G(A) \in \{0,1\}^{M \times N}$ and $G'(A) \in \{0,1\}^{M' \times N'}$ are isomorphic if and only if $M = M'$, $N = N'$, and*

1. *There exist bijections[6] $f_F : [M] \to [M]$ and $f_V : [N] \to [N]$ such that $G(A_{ai}) = G'(A_{bj})$ for all $a \in [M]$ and $i \in [N]$, where $b = f_F(a)$ and $j = f_V(i)$.*

2. *There exists a bijection for every factor,*

$$f_a^{idx} : \{1, \ldots, |G(F_a^{idx})|\} \to \{1, \ldots, |G'(F_b^{idx})|\} \quad \forall a \in [M], \tag{6}$$

*such that $f_V\big(G(F_{ak}^{idx})\big) = G'(F_{bl}^{idx})$ and $G(F_a^p) = \sigma_a\big(G'(F_b^p)\big)$, where where $b = f_F(a)$, $l = f_a^{idx}(k)$, $\sigma_a = \big((f_a^{idx}(1), f_a^{idx}(2), \ldots, f_a^{idx}(|G(F_a^{idx})|)\big)$, and $\sigma_a\big(G'(F_b^p)\big)$ denotes permuting the dimensions of the tensor $G'(F_b^p)$ according to $\sigma_a$.*

Condition 1 in Definition 1 states that permuting the *global* indices of variables or factors in a factor graph results in an isomorphic factor graph. Condition 2 in Definition 1 states that permuting the *local* indices of variables within factors also results in an isomorphic factor graph. In Lemmas 1, 2, and 3 we formalize the equivariance of messages and beliefs obtained by applying BPNN iterative layers. We use using the bijections from Definition 1 to construct bijective mappings between messages and beliefs. In Lemma 4 we use the equivariance of beliefs between isomorphic factor graphs to show that the output of BPNN-B is identical for isomorphic factor graphs.

**Lemma 1.** *Message equivariance: Let $g_{i \to a}^{(k)}$ and $h_{i \to a}^{(k)}$ denote variable to factor messages and $g_{a \to i}^{(k)}$ and $h_{a \to i}^{(k)}$ factor to variable messages obtained by applying k iterations of BP to factor graphs $G$ and $G'$. If $G$ and $G'$ are isomorphic as factor graphs and messages are initialized to a constant[7] then there is a bijective mapping between messages: $g_{i \to a}^{(k)} = h_{j \to b}^{(k)}$ and $g_{a \to i}^{(k)} = h_{b \to j}^{(k)}$ where $j = f_V(i)$ and $b = f_F(a)$. This property holds for BPNN-D iterative layers if $H(\cdot)$ is equivariant to global node indexing.*

*Proof.* We use a proof by induction.

Base case: the initial messages are all equal when constant initialization is used and therefore satisfy any bijective mapping.

Inductive step: Writing the definition of variable to factor messages, we have

$$g_{i \to a}^{(k)}(x_i) = \prod_{c \in \mathcal{N}(i) \setminus a} g_{c \to i}^{(k-1)}(x_i) = \prod_{c \in \mathcal{N}(j) \setminus b} h_{c \to j}^{(k-1)}(x_j) = h_{j \to b}^{(k)}(x_j), \tag{7}$$

since the bijective mapping holds for factor to variable messages at iteration $k-1$ by the inductive hypothesis. Writing the definition of factor to variable messages, we have

$$
\begin{aligned}
g_{a \to i}^{(k)}(x_i) &= \sum_{\mathbf{x}_a \backslash x_i} G(F_a^p)(\mathbf{x}_a) \prod_{l \in \mathcal{N}(a) \backslash i} g_{l \to a}^{(k)}(x_l) \\
&= \sum_{\mathbf{x}_b \backslash x_j} \sigma_a\big(G'(F_b^p)\big)(\mathbf{x}_b) \prod_{l \in \mathcal{N}(b) \backslash j} g_{l \to b}^{(k)}(x_l) = g_{b \to j}^{(k)}(x_j).
\end{aligned}
\tag{8}
$$

showing that the bijective mapping continues to hold at iteration $k$.

Proof extension to BPNN-D: the logic of the proof is unchanged when BP is performed in log-space with damping. The only difference between BPNN-D and standard BP is the replacement of the term $\alpha\big(\overline{m}_{a \to i}^{(k-1)} - \tilde{m}_{a \to i}^{(k)}\big)$ in the computation of factor to variable messages with $\Delta_{a \to i}^{(k)}$, where $\Delta^{(k)} = H\big(\overline{n}^{(k-1)} - \tilde{n}^{(k)}\big)$. If $H(\cdot)$ is equivariant to global node indexing (the bijective mapping $\Delta_{a \to i}^{(k)}(G) = \Delta_{b \to j}^{(k)}(G')$ holds, where $\Delta_{a \to i}^{(k)}(G)$ denotes applying the operator $H(\cdot)$ to the k-th iteration's message differences when the input factor graph is $G$ and taking the output corresponding to message $a \to i$), then equality is maintained in Equation 8 and the bijective mapping between messages holds. $\qquad\square$

**Lemma 2.** *Variable belief equivariance: Let $g_i^{(k)}$ and $h_i^{(k)}$ denote the variable beliefs obtained by applying k iterations of BP (or BPNN-D iterative layers with $H(\cdot)$ equivariant to global node indexing) to factor graphs $G$ and $G'$. If $G$ and $G'$ are isomorphic as factor graphs, then there is a bijective mapping between beliefs: $g_i^{(k)} = h_j^{(k)}$, where $j = f_V(i)$.*

*Proof.* By the definition of variable beliefs,

$$
g_i^{(k)}(x_i) = \frac{1}{z_i} \prod_{a \in \mathcal{N}(i)} g_{a \to i}^{(k)}(x_i) = \frac{1}{z_j} \prod_{a \in \mathcal{N}(j)} h_{a \to j}^{(k)}(x_j) = h_j^{(k)}(x_j),
\tag{9}
$$

where the second equality holds due to factor to variable message equivariance from Lemma 1. $\quad\square$

**Lemma 3.** *Factor belief equivariance: Let $g_a^{(k)}$ and $h_a^{(k)}$ denote the factor beliefs obtained by applying k iterations of BP (or BPNN-D iterative layers with $H(\cdot)$ equivariant to global node indexing) to factor graphs $G$ and $G'$. If $G$ and $G'$ are isomorphic as factor graphs, then there is a bijective mapping between beliefs: $g_a^{(k)} = \sigma_a\big(h_b^{(k)}\big)$, where $b = f_F(a)$ and $\sigma_a = \big((f_a^{idx}(1), f_a^{idx}(2), \ldots, f_a^{idx}(|G(F_a^{idx})|)\big)$.*

*Proof.* By the definition of factor beliefs,

$$
g_a^{(k)}(\mathbf{x}_a) = \frac{G(F_a^p)(\mathbf{x}_a)}{z_a} \prod_{i \in \mathcal{N}(a)} g_{i \to a}^{(k)}(x_i) = \frac{\sigma_a\big(G'(F_b^p)\big)(\mathbf{x}_b)}{z_b} \prod_{i \in \mathcal{N}(b)} h_{i \to b}^{(k)}(x_i) = \sigma_a\big(h_b^{(k)}(\mathbf{x}_b)\big),
\tag{10}
$$

where the second equality holds due to variable to factor message equivariance from Lemma 1. $\quad\square$

**Lemma 4.** *Bethe approximation invariance: If factor graphs $G$ and $G'$ are isomorphic, then the Bethe approximations obtained by applying BP to $G$ and $G'$ (or the output of BPNN-B) are identical.*

*Proof.* By the definition of the Bethe approximation (or the negative Bethe free energy),

$$
\begin{aligned}
-F_{\text{Bethe}}(G) &= \sum_{a=1}^{M} \sum_{\mathbf{x}_a} g_a(\mathbf{x}_a) \ln G(F_a^p)(\mathbf{x}_a) \\
&\quad - \sum_{a=1}^{M} \sum_{\mathbf{x}_a} g_a(\mathbf{x}_a) \ln g_a(\mathbf{x}_a) + \sum_{i=1}^{N} (d_i - 1) \sum_{x_i} g_i(x_i) \ln g_i(x_i) \\
&= \sum_{b=1}^{M} \sum_{\mathbf{x}_b} \sigma_{a'}(h_b)(\mathbf{x}_b) \ln \sigma_{a'}\big(G'(F_b^p)\big)(\mathbf{x}_b) \\
&\quad - \sum_{b=1}^{M} \sum_{\mathbf{x}_b} \sigma_{a'}(h_b)(\mathbf{x}_b) \ln \sigma_{a'}(h_b)(\mathbf{x}_b) + \sum_{j=1}^{N} (d_j - 1) \sum_{x_j} h_j(x_j) \ln h_j(x_j) \\
&= -F_{\text{Bethe}}(G')
\end{aligned}
\tag{11}
$$

where $a' = f_F^{-1}(b)$, the second equality follows from the equivariance of variable and factor beliefs (Lemmas 2 and 3), and the final equality follows from the commutative property of addition.

Proof extension to BPNN-B: the proof holds for BPNN-B because every permutation (in $S_{|\mathbf{x}_a|}$) of factor belief terms is input to $\text{MLP}_{BF}$. $\square$

## B  Extended Background

We provide background on belief propagation and graph neural networks (GNN) to motivate and clarify belief propagation neural networks (BPNN).

### B.1  BELIEF PROPAGATION

We describe a general version of belief propagation [52] that operates on factor graphs.

**Factor Graphs.**  A factor graph [32, 52] is a general representation of a distribution over $n$ discrete random variables, $\{X_1, X_2, \ldots, X_n\}$. Let $x_i$ denote a possible state of the $i^{th}$ variable. We use the shorthand $p(\mathbf{x}) = p(X_1 = x_1, \ldots, X_n = x_1)$ for the joint probability mass function, where $\mathbf{x} = \{x_1, x_2, \ldots, x_n\}$ is a specific realization of all $n$ variables. Without loss of generality, $p(\mathbf{x})$ can be written as the product

$$
p(\mathbf{x}) = \frac{1}{Z} \prod_{a=1}^{M} f_a(\mathbf{x}_a).
\tag{12}
$$

The functions $f_1, f_2, \ldots, f_m$ each take some subset of variables as arguments; function $f_a$ takes $\mathbf{x_a} \subset \{x_1, x_2, \ldots, x_n\}$. We require that all functions are non-negative and finite. This makes $p(\mathbf{x})$ a well defined probability distribution after normalizing by the distribution's partition function

$$
Z = \sum_{\mathbf{x}} \left( \prod_{a=1}^{M} f_a(\mathbf{x}_a) \right).
\tag{13}
$$

A factor graph is a bipartite graph that expresses the factorization of the distribution in equation 12. A factor graph's nodes represent the $n$ variables and $M$ functions present in equation 12. The nodes corresponding to functions are referred to as factor nodes. Edges exist between factor nodes and variables nodes if and only if the variable is an argument to the corresponding function.

**Message Updates.**  Belief propagation performs iterative message passing. The message $m_{i \to a}^{(k)}(x_i)$ from variable node $i$ to factor node $a$ during iteration $k$ is a vector over the states of variable $x_i$, which can be viewed as containing the relative probabilities that $x_i$ is in each state based on information from all factors connected to $x_i$ except for factor $a$. It is computed according to the rule

$$
m_{i \to a}^{(k)}(x_i) \coloneqq \prod_{c \in \mathcal{N}(i) \setminus a} m_{c \to i}^{(k-1)}(x_i).
\tag{14}
$$

The message $m_{a \to i}^{(k)}(x_i)$ from factor node $a$ to variable node $i$ during iteration $k$ is also a vector over the states of variable $x_i$, which can be viewed as containing the relative probabilities that $x_i$ is in each state based on information available to factor $a$. It is computed according to the rule

$$m_{a \to i}^{(k)}(x_i) := \sum_{\mathbf{x}_a \setminus x_i} f_a(\mathbf{x}_a) \prod_{j \in \mathcal{N}(a) \setminus i} m_{j \to a}^{(k)}(x_j). \tag{15}$$

Messages are typically initialized either randomly or as constants. The BP algorithm estimates approximate marginal probabilities for each variable, referred to as beliefs. We denote the belief at variable node $i$, after message passing iteration $k$ is complete, as $b_i^{(k)}(x_i)$ which is computed as

$$b_i^{(k)}(x_i) = \frac{1}{z_i} \prod_{a \in \mathcal{N}(i)} m_{a \to i}^{(k)}(x_i), \text{ with normalization } z_i = \sum_{x_i} \prod_{a \in \mathcal{N}(i)} m_{a \to i}^{(k)}(x_i). \tag{16}$$

Similarly, BP computes joint beliefs over the sets of variables $\mathbf{x}_a$ associated with each factor $f_a$. We denote the belief over variables $\mathbf{x}_a$, after message passing iteration $k$ is complete, as $b_a^{(k)}(\mathbf{x}_a)$ which is computed as

$$b_a^{(k)}(\mathbf{x}_a) = \frac{f_a(\mathbf{x}_a)}{z_a} \prod_{i \in \mathcal{N}(a)} m_{i \to a}^{(k)}(x_i), \text{ with normalization } z_a = \sum_{\mathbf{x}_a} f_a(\mathbf{x}_a) \prod_{i \in \mathcal{N}(a)} m_{i \to a}^{(k)}(x_i). \tag{17}$$

**Partition Function Approximation.** The belief propagation algorithm proceeds by iteratively updating variable to factor messages (Equation 14) and factor to variable messages (Equation 15) until they converge to fixed values, referred to as a fixed point of Equations 14 and 15, or a predefined maximum number of iterations is reached. While BP is not guaranteed to converge in general, whenever a fixed point is found it defines a set of consistent beliefs, meaning that marginal beliefs at factor nodes agree with beliefs every variable node they are connected to. At this point the beliefs are used to compute a variational approximation of the factor graph's partition function. This approximation, originally developed in statistical physics, is known as the Bethe free energy $F_{\text{Bethe}} \approx -\ln Z$ [10]. It is defined in terms of the Bethe average energy $U_{\text{Bethe}}$ and the Bethe entropy $H_{\text{Bethe}}$.

**Definition 2.** $U_{Bethe} := -\sum_{a=1}^{M} \sum_{\mathbf{x}_a} b_a(\mathbf{x}_a) \ln f_a(\mathbf{x}_a)$ *defines the Bethe average energy.*

**Definition 3.** $H_{Bethe} := -\sum_{a=1}^{M} \sum_{\mathbf{x}_a} b_a(\mathbf{x}_a) \ln b_a(\mathbf{x}_a) + \sum_{i=1}^{N} (d_i - 1) \sum_{x_i} b_i(x_i) \ln b_i(x_i)$ *defines the Bethe entropy, where $d_i$ is the degree of variable node $i$.*

**Definition 4.** *The Bethe free energy is defined as* $F_{Bethe} = U_{Bethe} - H_{Bethe}$.

**Numerically Stable Belief Propagation.** For numerical stability, belief propagation is generally performed in log-space and messages are normalized at every iteration. It is also standard to add a *damping* parameter, $\alpha \in [0, 1)$, to improve convergence by taking partial update steps. BP without damping is recovered when $\alpha = 0$, while $\alpha = 1$ would correspond to not updating messages and instead retaining their values from the previous iteration. With these modifications, the variable to factor messages from Equation 2 are rewritten as follows, where terms scaled by $\alpha$ represent the difference in the message's value from the previous iteration:

$$\overline{m}_{i \to a}^{(k)} = \tilde{m}_{i \to a}^{(k)} + \alpha \left( \overline{m}_{i \to a}^{(k-1)} - \tilde{m}_{i \to a}^{(k)} \right), \text{ where } \tilde{m}_{i \to a}^{(k)} = -z_{i \to a} + \sum_{c \in \mathcal{N}(i) \setminus a} \overline{m}_{c \to i}^{(k-1)}. \tag{18}$$

Similarly, the factor to variable messages from Equation 2 are rewritten as

$$\overline{m}_{a \to i}^{(k)} = \tilde{m}_{a \to i}^{(k)} + \alpha \left( \overline{m}_{a \to i}^{(k-1)} - \tilde{m}_{a \to i}^{(k)} \right), \ \tilde{m}_{a \to i}^{(k)} = -z_{a \to i} + \underset{\mathbf{x}_a \setminus x_i}{\text{LSE}} \left( \phi_a(\mathbf{x}_a) + \sum_{j \in \mathcal{N}(a) \setminus i} \overline{m}_{j \to a}^{(k)} \right), \tag{19}$$

Note that $\overline{m}_{i \to a}^{(k)}$ and $\overline{m}_{a \to i}^{(k)}$ are vectors of length $|X_i|$, $\phi_a(\mathbf{x}_a) = \ln(f_a(\mathbf{x}_a))$ denotes log factors, $z_{i \to a}$ and $z_{a \to i}$ are normalization terms, and we use the shorthand LSE for the log-sum-exp function:

$\underset{\mathbf{x}_a \setminus x_i}{\text{LSE}} \left( \phi_a(\mathbf{x}_a) \right) = \ln \left( \sum_{\mathbf{x}_a \setminus x_i} \exp \left( \phi_a(\mathbf{x}_a) \right) \right).$

Figure 5: Computation graph of BPNN iterative layers followed by BPNN-B.

## B.2   GNN Background

This section provides background on graph neural networks (GNNs), a form of neural network used to perform representation learning on graph structured data. GNNs perform iterative message passing operations between neighboring nodes in graphs, updating the learned, hidden representation of each node after every iteration. Xu et al. [51] showed that graph neural networks are at most as powerful as the Weisfeiler-Lehman graph isomorphism test [49], which is a strong test that generally works well for discriminating between graphs. Additionally, [51] presented a GNN architecture called the Graph Isomorphism Network (GIN), which they showed has discriminative power equal to that of the Weisfeiler-Lehman test and thus strong representational power. We will use GIN as a baseline GNN for comparison in our experiments because it is provably as discriminative as any GNN that aggregates information from 1-hop neighbors.

We now describe in detail the GIN architecture that we use as a baseline. Our architecture performs regression on graphs, learning a function $f_{\text{GIN}} : \mathcal{G} \to \mathbb{R}$ from graphs to a real number. Our input is a graph $G = (V, E) \in \mathcal{G}$ with node feature vectors $\mathbf{h}_v^{(0)}$ for $v \in V$ and edge feature vectors $\mathbf{e}_{u,v}$ for $(u, v) \in E$. Our output is the number $f_{\text{GIN}}(G)$, which should ideally be close to the ground truth value $y_G$. Let $\mathbf{h}_v^{(k)}$ denote the representation vector corresponding to node $v$ after the $k^{th}$ message passing operation. We use a slightly modified GIN update to account for edge features as follows:

$$\mathbf{h}_v^{(k)} = \text{MLP}_1^{(k)} \left( \mathbf{h}_v^{(k-1)} + \sum_{u \in \mathcal{N}(v)} \text{MLP}_2^{(k)} \left( \mathbf{h}_u^{(k-1)}, \mathbf{e}_{u,v} \right) \right). \tag{20}$$

A $K$-layer GIN network with width $M$ is defined by $K$ successive GIN updates as given by Equation 20, where $\mathbf{h}_v^{(k)} \in \mathbb{R}^M$ is an $M$-dimensional feature vector for $k \in \{1, 2, \ldots, K\}$. All MLPs within GIN updates (except $\text{MLP}_2^{(0)}$) are multilayer perceptrons with a single hidden layer whose input, hidden, and output layers all have dimensionality $M$. $\text{MLP}_2^{(0)}$ is different in that its input dimensionality is given by the dimensionality of the original node feature representations. The final output of our GIN network is given by

$$f_{\text{GIN}}(G) = \text{MLP}^{(K+1)} \left( \underset{k=1}{\overset{K}{\text{CONCAT}}} \sum_{v \in G} \mathbf{h}_v^k \right), \tag{21}$$

where we concatenate summed node feature vectors from all layers and $\text{MLP}^{(K+1)}$ is a multilayer perceptron with a single hidden layer. Its input and hidden layers have dimensionality $M \cdot K$ and its output layer has dimensionality 1.

## C   BPNN Iterative Layer Implementation Details

We implemented BPNN iterative layers to operate in log space with message normalization. In this section we write out the equations for BPNN with these modifications. BPNN iterative layers are flexible neural operators that can operate on beliefs or message in a variety of ways. The BPNN iterative damping layer (BPNN-D, shown in Fig 1) modifies factor-to-variable messages (Equation 19)

using the output of a learned operator $H : \mathbb{R}^{\sum_{i=1}^{n} d_i |X_i|} \to \mathbb{R}^{\sum_{i=1}^{n} d_i |X_i|}$ in place of the conventional damping term $\alpha\big(\overline{m}_{a\to i}^{(k-1)} - \tilde{m}_{a\to i}^{(k)}\big)$, where $d_i$ denotes the degree and $|X_i|$ the cardinality of variable $X_i$. This learned operator $H(\cdot)$ takes as input the difference between iterations $k-1$ and $k$ of every factor-to-variable message, and modifies these differences jointly. It can thus be much richer than a scalar multiplier. BPNN-D factor-to-variable messages are given by

$$\overline{n}_{a\to i}^{(k)} = \tilde{n}_{a\to i}^{(k)} + \Delta_{a\to i}^{(k)}, \; \tilde{n}_{a\to i}^{(k)} = -z_{a\to i} + \operatorname*{LSE}_{\mathbf{x}_a \backslash x_i} \left( \phi_a(\mathbf{x}_a) + \sum_{j \in \mathcal{N}(a) \backslash i} \overline{n}_{j\to a}^{(k)} \right), \tag{22}$$

Let $\Delta_{\mathbf{f}\to\mathbf{x}}^{(k)} = H\big(\overline{n}_{\mathbf{f}\to\mathbf{x}}^{(k-1)} - \tilde{n}_{\mathbf{f}\to\mathbf{x}}^{(k)}\big)$ denote the result of applying $H(\cdot)$ to all factor-to-variable message differences.[8] Then $\Delta_{a\to i}^{(k)}$ is the output corresponding to the modified $a \to i$ message difference.

Variable-to-factor messages are unchanged from Eq. 18, except for taking messages $\overline{n}_{a\to i}^{(k)}$ as input,

$$n_{i\to a}^{(k)} = \tilde{n}_{i\to a}^{(k)} + \alpha(n_{i\to a}^{(k-1)} - \tilde{n}_{i\to a}^{(k)}), \; \text{where } \tilde{n}_{i\to a}^{(k)} = -z_{i\to a} + \sum_{c \in \mathcal{N}(i) \backslash a} \overline{n}_{c\to i}^{(k-1)}. \tag{23}$$

Note that we recover Equations 18 and 19 exactly if $H$ is an elementwise function $H(x) = \alpha x$.

## C.1  Variants

When the convergence properties of BPNN-D are not needed (e.g., if BP is not a lower bound to the partition function of a particular problem), we have the flexibility to create BPNN iterative layers that directly operate on a combination of messages and beliefs by modifying $\tilde{m}_{i\to a}^{(k)}$ and $\tilde{m}_{a\to i}^{(k)}$ from Equations 18 and 19.

We can introduce a variant that parameterizes both factor to variable messages and factor beliefs and computes factor to variable messages as:

$$\tilde{m}_{a\to i}^{(k)} = -z_{a\to i} + \operatorname*{LSE}_{\mathbf{x}_a \backslash x_i} \left( \phi_a(\mathbf{x}_a) + \operatorname{LNE}_2 \left[ \sum_{j \in \mathcal{N}(a) \backslash i} \operatorname{LNE}_1 \left( \overline{m}_{j\to a}^{(k)} \right) \right] \right) \tag{24}$$

where we use the shorthand

$$\operatorname{LNE}_i(\mathbf{h}) = \ln\left( \operatorname{MLP}_{\theta_i}\big( \exp(\mathbf{h}) \big) \right), \tag{25}$$

and $\operatorname{MLP}_{\theta_i}$ is a multilayer perceptron parameterized by $\theta_i$. We exponentiate before applying the multilayer perceptron because we empirically find that this improves training as opposed to having MLPs operate directly in log space.

We can also introduce additional variants that operate only on messages. This is beneficial because it preserves equivariance under factor graph isomorphism, which applying MLPs to factor beliefs violates. We parameterize both variable to factor and factor to variable messages:

$$\tilde{m}_{i\to a}^{(k)} = -z_{i\to a} + \sum_{c \in \mathcal{N}(i) \backslash a} LNE_3(\overline{m}_{c\to i}^{(k-1)}). \tag{26}$$

$$\tilde{m}_{a\to i}^{(k)} = -z_{a\to i} + \operatorname*{LSE}_{\mathbf{x}_a \backslash x_i} \left( \phi_a(\mathbf{x}_a) + \sum_{j \in \mathcal{N}(a) \backslash i} LNE_4(\overline{m}_{j\to a}^{(k)}) \right), \tag{27}$$

BPNN iterative layers allow for great flexibility and different combinations of these MLPs can be applied in a specific layer depending on the task at hand. These MLPs can even be combined with the damping MLPs found in BPNN-D layers in lieu of the fixed scalar damping coefficient $\alpha$ found in Equations 18 and 19.

**BPNN Initialization** Note that any BPNN architecture built from iterative layers with or without a BPNN-B layer can be initialized to perform BP run for a fixed number of iterations by initializing MLPs functions $f(x) = x$. E.g. weight matrices are set to the identity, bias terms to zero, and any nonlinearities are chosen so as to avoid affecting the input at initialization.

# D Ising Model Experiments

**Data Generation.** An $N \times N$ Ising model is defined over binary variables $x_i \in \{-1, 1\}$ for $i = 1, 2, \ldots, N^2$, where each variable represents a spin. Each spin has a local field parameter $J_i$ which corresponds to its local potential function $J_i(x_i) = J_i x_i$. Each spin variable has 4 neighbors, unless it occupies a grid edge. Neighboring spins interact with coupling potentials $J_{i,j}(x_i, x_j) = J_{i,j} x_i x_j$. The probability of a complete variable configuration $\mathbf{x} = \{x_1, \ldots, x_{N^2}\}$ is defined to be

$$p(\mathbf{x}) = \frac{1}{Z} \exp\left(\sum_{i \in V} J_i x_i + \sum_{(i,j) \in E} J_{i,j} x_i x_j\right), \qquad (28)$$

where the normalization constant $Z$, or partition function, is defined to be

$$Z = \sum_{\mathbf{x}} \exp\left(\sum_{i \in V} J_i x_i + \sum_{(i,j) \in E} J_{i,j} x_i x_j\right). \qquad (29)$$

We performed experiments using datasets of randomly generated Ising models. Each dataset was created by first choosing $N$, $c_{\max}$, and $f_{\max}$. We sampled $N \times N$ Ising models according to the following process

$$c \sim \text{Unif}[0, c_{\max}),$$
$$f \sim \text{Unif}[0, f_{\max}),$$
$$(J_i)_{i \in V} \overset{\text{i.i.d.}}{\sim} \text{Unif}[-f, f),$$
$$(J_{i,j})_{(i,j) \in E} \overset{\text{i.i.d.}}{\sim} \text{Unif}[0, c).$$

**Baselines.** We trained a 10 layer GNN (GIN architecture) with width 4 on the same dataset of attractive Ising models that we used for our BPNN. We set edge features to the coupling potentials; that is, $\mathbf{e}_{u,v} = J_{u,v}$. We set the initial node representations to the local field potentials of each node, $\mathbf{h}_v^{(0)} = J_v$. We used the same training loss and optimizer as for our BPNN. We used an initial learning rate of 0.001 and trained for 5k epochs, decaying the learning rate by .5 every 2k epochs.

We consider two additional baselines: Bethe approximation from running standard loopy belief propagation and mean field approximation. We used the libDAI [35] implementation for both. We test loopy belief propagation with and without damping and with both parallel and sequential message update strategies. We show results for two settings whose estimates of the partition function differ most drastically: (1) run for a maximum of 10 iterations with parallel updates and damping set to .5, and (2) run for a maximum of 1000 iterations with sequential updates using a random sequence and no damping.

**Improved Lower Bounds and Faster Convergence.** We trained a BPNN-D to estimate the partition function on a training set of 50 random Ising models. We randomly sampled the number of iterations of BPNN-D to apply during training between 5 and 30. When BPNN-D is then run to convergence on a validation set of random Ising models, we find that (1) it finds fixed points that provide tighter lower bounds on the partition function as explained in the main text and (2) it converges faster than BP as shown in Figure 6.

**Out of Distribution Generalization.** We tested BPNN's ability to generalize to larger factor graphs and to shifts in the test distribution. Again we used a training set of 50 Ising models. We sampled test data from distributions with $c_{max}$ and $f_{max}$ increased by factors of 2 and 10 from their training values,

Figure 6: The maximum difference in factor to variable message values between iterations is plotted against the message passing iteration for BPNN-D and BPNN run on 50 validation Ising models. BPNN-D converges to a maximum difference of $10^{-5}$ faster than BP, with a median speedup of 1.7x.

with $N$ set to 14 (for 196 variables instead of the 100 seen during training). For this experiment we used a BPNN architecture with 10 iterative layers whose weights were not tied and with MLPs that operate on factor messages. For the out of distribution experiments, we did not use a final BPNN-B layer, we set the residual parameters to $\alpha_0 = \alpha_1 = \alpha_2 = .5$, and trained on 50 attractive Ising models generated with $N = 10$, $f_{\max} = .1$, and $c_{\max} = 5$. We used mean squared error as our training loss. We used the Adam optimizer [29] with an initial learning rate of .0005 and trained for 100 epochs, with a decay of .5 after 50 epochs. Batching was over the entire training set (of size 50) with one optimization step per epoch.

# E  SAT Experiments

**Additional Dataset Details**  We evaluated the performance of our BPNN using the suite of benchmarks from Soos and Meel [44]. Some of these benchmarks come with a sampling set. The sampling set redefines the model counting problem, asking how many configurations of variables in the sampling set correspond to at least one complete variable configuration that satisfies the formula. (A formula with $n$ variables may have at most $2^n$ satisfying solutions, but a sampling set over $i$ variables will restrict the number of solutions to at most $2^i$). We stripped all problems of sampling sets since they are outside the scope of this work. We also stripped all problems of minimal independent support variables sets and recomputed these when possible (we will discuss further later in this section).We ran the exact model counter DSharp[9] [37] on all benchmarks with a timeout of 5k seconds to obtain ground truth model counts for 928 of the 1,896 benchmarks. Only 50 of these problems had more than 5 variables in the largest factor, so we discarded these problems and set the BPNN architecture to run on factors over 5 variables. We categorized the remaining 878 by their arcane names into groupings. With some sleuthing we determined that categories 'or_50', 'or_60', 'or_70', and 'or_100' contain network DQMR problems with 150, 121, 111, and 138 benchmarks per category respectively. Categories '75' and '90' contain network grid problems with 20 and 107 benchmarks per category respectively. Category 'blasted' conains bit-blasted versions of SMTLIB ( satisfiability modulo theories library) benchmarks [12] and has 147 benchmarks. Category 's' contains representations of circuits with a subset of outputs randomly xor-ed and has 68 benchmarks. We discarded 4 categories that contained fewer than 10 benchmarks. For each category that contained more than 10 benchmarks, we split 70% into the training set and left the remaining benchmarks in the test set. We then performed two splits of the training set for training and validation; for each category we (1) trained on a random sampling of 70% of the training problems and performed validation on the remaining 30% and (2) trained on 70% of the training problems that DSharp solved fastest and performed validation on the remaining 30% that took longest for DSharp to solve. These hard validation sets are significantly

more challenging for Dsharp. The median runtime in each category's hard validation set is 4 to 15 times longer than the longest runtime in each corresponding easy training set.

**Minimal Independent Support**    As a pre-processing step for ApproxMC3 and F2, we attempted to find a set of variables that define a *minimal independent support* (MIS) [25] for each benchmark using the authors' code[10] with a timeout of 1k seconds. A set of variables that define a MIS for a boolean formula fully determine the values of the remaining variables. Randomized hashing algorithms can run significantly faster when given a set of variables that define a MIS. When we could find a set of variables that define a MIS, we recorded the time that each randomized hashing algorithm required without the MIS and the sum of the time to find the MIS and perform randomized hashing with the MIS. We report the minimum of these two times.

**Baseline Approximate Model Counters.**    For comparison, we ran the state of the art approximate model counter ApproxMC3[11] [12, 44] on all benchmarks. ApproxMC3 is a randomized hashing algorithm that returns an estimate of the model count that is guaranteed to be within a multiplicative factor of the exact model count with high probability. Improving the guarantee, either by tightening the multiplicative factor or increasing the confidence, will increase the algorithm's runtime. We ran ApproxMC3 with the default parameters; confidence set to 0.81 and epsilon set to 16.

We also compare with the state of the art randomized hashing algorithm F2[12] from [4, 5], run with CryptoMiniSat5[13] [45, 44]. This algorithm gives up the probabilistic guarantee that the returned estimate will be within a multiplicative factor of the true model count in return for significantly increased computational efficiency. We computed only a lower bound and ran F2 with variables appearing in only 3 clauses. This significantly speeds up the reported results [4, p.14], at some additional cost to accuracy. For example, on the problem 'blasted_case37' [4, p.14] report an estimate of $\log_2(\#\text{models}) \approx 151.02$ and a runtime of 4149.9 seconds. Running F2 with variables appearing in only 3 clauses, we computed the lower bound on $\log_2(\#\text{models})$ of 148 in 2 seconds.

We also attempted to train a GNN, using the architecture from [43] to perform regression instead of classification. We used the author's code, making slight modifications to perform regression. However, we were not successful in achieving non-trivial learning.

**BPNN Training Protocol.**    We trained our BPNN to predict the natural logarithm of the number of satisfying solutions to an input boolean formula. We consider the general case of an input formula over $n$ boolean variables, $\{X_1, X_2, \ldots, X_n\}$, in conjunctive normal form (CNF). Formulas in CNF are a conjunction of clauses, where each clause is a disjunction of literals. A literal is either a variable or its negation. We converted boolean formulas into factor graphs where each clause corresponds to a factor. Factors take the value of 1 for variable configurations that satisfy the clause and 0 for variable configurations that do not satisfy the clause. The partition function of this factor graph is equal to the number of satisfying solutions. We trained a BPNN architecture composed of 5 BPNN-D layers followed by a BPNN-B layer. We used the Adam optimizer [29] with learning rate decay.

**Ablation Study.**    The columns labeled BPNN-NI and BPNN-DC in table 2 correspond to ablated versions of our BPNN model. We trained a BPNN with a BPNN-B layer that was not permutation invariant to local variable indexing, by removing the sum over permutations in $S_{|\mathbf{x}_a|}$ from Equation 5 and only passing in the original beliefs. We refer to this non-invariant version as BPNN-NI. We then forced BPNN-NI to 'double count' messages by changing the sums in Equations 22 and 23 to be over $j \in \mathcal{N}(a)$. We this non-invariant version that performs double counting as BPNN-DC. We observe validation improvement in BPNN over these ablated versions when generalization is particularly challenging, e.g. on 'blasted' problems individually and on all categories.

**Additional Baseline Approximate Model Counter Information**    Table 3 shows the root mean squared error (RMSE) of estimates from the approximate model counters ApproxMC3 and F2 across all training benchmarks in each category. Error was computed as the difference between the natural logarithm of the number of satisfying solutions and the estimate. The fraction of benchmarks that

| RMSE Ablation Study by SAT Category | | | | |
|---|---|---|---|---|
| Benchmark Category | Train / Val Split | Train / Val BPNN | Train / Val BPNN-NI | Train / Val BPNN-DC |
| 'or_50' | Random Split | .32 / **.30** | 0.18 / 0.37 | 0.95 / 2.21 |
| | Easy / Hard | .31 / 1.10 | 0.17 / **0.70** | 0.72 / 4.81 |
| 'or_60' | Random Split | .32 / **.39** | 0.20 / 0.43 | 0.70 / 2.36 |
| | Easy / Hard | .31 / 1.7 | 0.21 / **1.22** | 0.57 / 3.69 |
| 'or_70' | Random Split | .28 / .53 | 0.19 / **0.45** | 0.79 / 1.85 |
| | Easy / Hard | .35 / **.50** | 0.19 / 0.58 | 0.73 / 2.75 |
| 'or_100' | Random Split | .48 / **.51** | 0.31 / 0.59 | 1.05 / 2.37 |
| | Easy / Hard | .48 / **.58** | 0.27 / 0.60 | 0.89 / 288.91 |
| 'blasted' | Random Split | 4.39 / **4.24** | 4.22 / 10.39 | 3.01 / 6.59 |
| | Easy / Hard | 2.19 / 10.10 | 1.51 / **8.28** | 1.57 / 948.95 |
| '75' | Random Split | 1.69 / 1.39 | 1.31 / 1.34 | 0.66 / **0.36** |
| | Easy / Hard | 1.51 / 2.81 | 0.98 / 2.96 | 0.65 / **2.00** |
| '90' | Random Split | 2.46 / 2.18 | 1.59 / 2.21 | 0.94 / **1.59** |
| | Easy / Hard | 2.17 / 2.86 | 1.76 / 3.27 | 0.81 / **1.23** |
| All Categories | Random Split | 3.92 / **5.31** | 6.77 / 9.57 | 4.28 / 27.03 |

Table 2: RMSE of BPNN for each training/validation set, along with ablation results. BPNN corresponds to a model with 5 BPNN-D layers followed by a Bethe layer that is invariant to the factor graph representation. BPNN-NI corresponds to removing invariance from the Bethe layer. BPNN-DC corresponds to performing 'double counting' as is standard for GNN, rather than subtracting previously sent messages as is standard for BP. 'Random Split' rows show that BPNNs are capable of learning a distribution from a tiny dataset of only 10s of training problems. 'Easy / Hard' rows additionally show that BPNNs are able to generalize from simple training problems to significantly more complex validation problems.

| Baselines RMSE by SAT Category | | |
|---|---|---|
| | RMSE (% Completed) | |
| Category | ApproxMC3 | F2 |
| 'or_50' | 0.07 (89%) | 2.4 (100%) |
| 'or_60' | 0.07 (87%) | 2.3 (100%) |
| 'or_70' | 0.06 (78%) | 2.4 (100%) |
| 'or_100' | 0.06 (73%) | 2.4 (100%) |
| 'blasted' | 0.04 (80%) | 2.4 (84%) |
| '75' | 0.04 (92%) | 2.0 (100%) |
| '90' | 0.03 (16%) | 12.4 (68%) |

Table 3: Root mean squared error (RMSE) of estimates of the natural logarithm of the number of satisfying solutions is shown. The fraction of benchmarks within each category that each approximate counter was able to complete within the time limit of 5k seconds is shown in parentheses.

each approximate counter was able to complete within the time limit of 5k seconds is also shown. For each benchmark category we show runtime percentiles for ApproxMC3, F2, and the exact model counter DSharp in Table 4. The DSharp runtime column shows the runtime dividing our easy training sets and hard validation sets for each benchmark category. It also shows the median run time of each hard validation set (85th percentile). The median runtime in each category's hard validation set is 4 to 15 times longer than the longest runtime in each corresponding easy training set. We observe that F2 is generally tens or hundreds of times faster than ApproxMC3. On these benchmarks DSharp is generally faster than F2, however there exist problems that can be solved much faster by randomized hashing (ApproxMC3 or F2) than by DSharp [5, 44].

|  | **Runtimes By Percentile** | | |
| Category | DSharp (0/70/85/100) | ApproxMC3 (0/70/100) | F2 (0/70/100) |
|---|---|---|---|
| 'or_50' | 0.0 / 0.8 / 12.4 / 48.1 | 0.1 / 336.6 / 5k | 0.2 / 4.0 / 89.9 |
| 'or_60' | 0.0 / 0.3 / 2.1 / 79.1 | 0.1 / 276.6 / 5k | 0.2 / 5.0 / 353.2 |
| 'or_70' | 0.0 / 0.7 / 3.6 / 46.6 | 0.1 / 748.3 / 5k | 0.2 / 11.9 / 491.3 |
| 'or_100' | 0.0 / 0.3 / 4.8 / 54.2 | 0.1 / 1918.8 / 5k | 0.2 / 33.0 / 3021.1 |
| 'blasted' | 0.0 / 1.7 / 29.3 / 1390.8 | 0.0 / 952.6 / 5k | 0.0 / 742.3 / 5k |
| '75' | 0.0 / 6.0 / 29.0 / 160.3 | 279.6 / 805.1 / 5k | 1.1 / 2.3 / 9.0 |
| '90' | 0.0 / 1.8 / 16.7 / 479.9 | 326.3 / 5k / 5k | 1.1 / 5k / 5k |

Table 4: Runtime percentiles (in seconds) are shown for DSharp, ApproxMC3, and F2. Percentiles are computed separately for each category's training dataset. In comparison, BPNN sequential runtime is nearly a constant and BPNN parallel runtime is limited by GPU memory.

# F  Stochastic Block Model Experiments

**Stochastic Block Model Definition**   A $C$ class Stochastic Block Model (SBM) is a randomly generated graph with $N$ vertices, class assignment probabilities $p_i; i \in 1, \ldots, C$, where $\sum_{i=1}^{C} p_i = 1$, and edge probabilities $e_{ij}; i, j \in 1, \ldots, C$. Then, to generate the graph, we sample a class for each node, $c_m; m \in 1, \ldots, N$ in accordance with the class assignment probabilities. Then, we sample the edge set $E$ in the following manner: we take every pair of nodes $x_m, x_n; m, n \in 1, \ldots, N$ and with probability $e_{c_m,c_n}$ assign an edge between those nodes.

**SBM Factor Graph Construction**   We can construct a factor graph for a sampled SBM model. This factor graph represents a function over arrangements of node class membership, which, when normalized, gives the joint probability distribution of observing those class memberships given the SBM generation parameters and sampled SBM structure. For a given SBM with $N$ nodes, $C$ classes, class assignment probabilities $p_i; i \in 1, \ldots, C$, sampled class assignments $c_m; m \in 1, \ldots, N$, edge probabilities $e_{ij}; i, j \in 1, .., C$, and sampled edge set $E$, we have the following unary factor potentials $f_i(x_m); i \in 1, \ldots, C$ for every node $x_m; m \in 1, \ldots, N$:

$$f_i(x_m) = p_i$$

We can construct binary factor potentials $f_{ij}(x_m, x_n); i, j \in 1, .., C$ between nodes $x_m, x_n; (m, n) \in E$ as:

$$f_{ij}(x_m, x_n) = e_{c_m,c_n}$$

and between nodes $x_m, x_n; (m, n) \notin E$ as:

$$f_{ij}(x_m, x_n) = 1 - e_{c_m,c_n}$$

Note that when we fix a variable to a specific value (which is needed to calculate marginals, as explained below), we simply set all factor potentials involving that variable that do not agree with that value to zero.

**Marginal Calculation from Log Partitions**   Training a model to estimate partition functions with fixed variables is advantageous as we train the model to perform tasks that can directly be used to compute marginals, which are the probabilities that a node belongs to a specific class. This can be used to perform community detection or to quantify uncertainty and rare events in community membership. To see how we compute marginals with our experimental setup, take a two class SBM, select a node $x_m$, fix its value to class 0 to obtain log partition function $\ln(Z_1)$ and then fix its value to class 1 to obtain log partition $\ln(Z_2)$. Then, the log marginals $\ln(x_m^0)$ and $\ln(x_m^1)$ are simply:

$$\ln(x_m^0) = \ln(Z_1) - \ln(Z_1 + Z_2)$$

and

$$\ln(x_m^1) = \ln(Z_2) - \ln(Z_1 + Z_2)$$

where

$$\ln(Z_1 + Z_2)$$

| | | **Out of Distribution SBM RMSE** | | | |
|---|---|---|---|---|---|
| Nodes | Data Edge Probs | Data Class Probs | BP RMSE | GNN RMSE | BPNN RMSE |
| 15 | (.93, .067) | (.6, .4) | 12.27 | 9.21 | **4.61** |
| 15 | (.93, .067) | (.8, .2) | 11.22 | 12.19 | **5.68** |
| 15 | (.967, .033) | (.6, .4) | 16.77 | 12.62 | **4.92** |
| 15 | (.9, .1) | (.8, .2) | 8.99 | 16.83 | **6.53** |
| 15 | (.967, .13) | (.75, .25) | 9.54 | 12.77 | **6.64** |
| 15 | (.867, .033) | (.75, .25) | 11.55 | 9.17 | **4.14** |
| 16 | (.9375, .0625) | (.75, .25) | 13.88 | 15.07 | **7.08** |
| 17 | (.94, .06) | (.75, .25 | 15.92 | 17.89 | **8.43** |
| 18 | (.94, .06) | (.75, .25) | 15.81 | 20.90 | **10.50** |
| 19 | (.95, .05) | (.75, .25) | 18.6 | 22.77 | **10.61** |
| 20 | (.95, .05) | (.75, .25) | 19.37 | 28.31 | **15.23** |

Table 5: RMSE of $\ln(Z)$ of BPNN against BP and GNN for SBM's generated from different distributions and larger graphs than the training or validation set. We see that BPNN outperforms both methods here across different edge probabilities, class probabilities, and on larger graphs. Furthermore, it generalizes better than GNN in all these settings.

can be computed in a numerically stable fashion from $\ln(Z_1)$ and $\ln(Z_2)$ using the logsumexp trick.

**Model and Training Details**   For baselines, we ran Belief Propagation to convergence with parallel updates and damping coefficient .5 as well as a Graph Isomorphism Network (GIN) with 30 layers and width 8. GIN is maximally discriminative among GNNs that consider 1-hop neighbors, which is computationally comparable to BPNN. In our evaluations, GIN performs comparably to more computationally expensive two hop GNNs on the related problem of SBM community detection [14]. We trained our GIN GNN architecture on the 5 class graph coloring community detection setting described in [14] and compared it to the performance of the two hop GNNs described there. Our GNN had 20 layers with a width of 8 and achieved a permutation invariant validation overlap score of .166 when trained for the same number of iterations, nearly identical to the two hop GNN performance reported in [14]. Since one hop GNNs train significantly faster than two hop, we managed to obtain overlap scores as high as .185 when training for longer. In any case, our one hop GNN performs comparably with two hop GNN architectures on the related task of SBM community detection and thus, in addition to its convenience, makes for a strong baseline method. For all models, we trained for 300 epochs on 1 GPU with an Adam Optimizer (learning rate of 2e-4, batch size of 8) minimizing Mean Squared Error between the estimated log partition and true log partition.

**Out of Distribution Generalization**   We test the capacity of our BPNN (with no double counting and an invariant BPNN-B layer) to generalize to out of distribution graphs compared to the GNN model, while comparing both against the BP benchmark. Since the factor graphs are fully connected, slight changes to the initial parameters can produce rather large differences in the graphs and their log partition function. In addition to perturbing the initial class probabilities and edge probabilities, we also test the ability of BPNN to generalize to larger graphs, which is a desirable property as the Junction Tree algorithm for exact inference becomes exponentially more expensive as the size of the graph grows due to the fully connected nature of SBM factor graphs. For each scenario, we generate five separate graphs and generate test examples as mentioned previously. We present our results in Table 5. We observe that BPNN performs the best of all three methods when class and edge probabilities are changed and generalizes better than GNN in these settings as well. Furthermore, when the size of graphs are increased, BPNN can outperform BP and GNN on graphs with as many as 20 nodes (a setting with over 80% more edges than training) and generalizes significantly better than GNN.

Since our SBM factor graphs are fully connected, adding $n$ times more nodes leads to a $O(n^2)$ increase in edges which may make it tougher for the model to generalize to larger and larger graphs. Using an auxiliary field approximation for SBM message passing, as described in [15] can help generalization to larger graphs, as in this case the increase in edges will increase linearly with graph size, and this is something to investigate further.

**Marginal Estimation**    We also compared BPNN to BP for marginal estimation, using the estimated log partition functions with single nodes set to a fixed value to calculate marginals for those nodes, as described above. Under the graph parameters used in these experiments, the marginals are usually extremely close to 1 and 0, but in such dense graphs, changes to the magnitude of these marginals can have large effects on the log partition function calculation. In some cases, BP computes the correct marginals under these conditions, but in some cases, it is off by 20-30 orders of magnitude on the smaller marginal. Such errors do not affect community recovery, however, when we care about very rare outcomes, they can have a big effect on quantifying uncertainty in community membership. On 15 node graphs, BPNN, by learning more accurate log partitions, is on average almost 5 orders of magnitude closer to the true marginals than BP and an order more accurate than GNN in distribution. Furthermore, BPNN outperforms BP when trained on graphs of 15 nodes and tested on graphs of as many as 20 nodes. While in this work we focus mainly on partition function estimation, as these estimates are better understood in the context of Loopy Belief Propagation, BPNN also shows promising results on estimating marginals. We believe these results could be further improved by training explicitly to estimate marginals, e.g. by correctly predicting the difference in partitions between graphs with one variable fixed to either value, and this is a promising avenue for further research.

## Footnotes

[4]Note that a factor graph can be viewed as a weighted hypergraph where factors define hyperedges and factor potentials define hyperedge weights for every variable assignment within the factor.

[5]For readability, we use $a$ and $b$ to index factors and $i$ and $j$ to index variables throughout this section.

[6]For $K \in \mathbb{N}$, we use $[K]$ to denote $\{1, 2, \ldots, K\}$.

[7]Any message initialization strategy can be used, as long as initial messages are equivariant; e.g. they satisfy the bijective mapping $g_{i \to a}^{(0)} = h_{j \to b}^{(0)}$ and $g_{a \to i}^{(k)} = h_{b \to j}^{(k)}$ where $j = f_V(i)$ and $b = f_F(a)$.

[8]We use message subscripts $\mathbf{f}$ and $\mathbf{x}$ to denote sets of messages, e.g., $\overline{n}_{\mathbf{f}\to\mathbf{x}}^{(k)} = \{\overline{n}_{a\to i}^{(k)} : f_a \in \mathbf{f}, x_i \in \mathbf{x}\}$.

[9]https://github.com/QuMuLab/dsharp

[10] https://github.com/meelgroup/mis

[11] https://github.com/meelgroup/ApproxMC

[12] https://github.com/ptheod/F2

[13] https://github.com/msoos/cryptominisat