[Reviews · NeurIPS 2020]

Review 1

Summary and Contributions: The authors propose belief propagation neural networks: a modification of graph neural networks that is designed to emulate (or improve upon) BP style message passing algorithms while trying to capture some of the same types of symmetries / structure that is present in the BP fixed points. The proposed approach is validated in a variety of different settings.

Strengths: - The approach of generalizing damped BP via non-constant damping is innovative as far as I am aware. The allowed functions, H, seem to be quite general even if the aim is simply to recover BP fixed points. - The experiments seem to show improved performance over BP and GNNs both on models generated from the training distribution and those that are slightly outside of it.

Weaknesses: - I found some of the discussion difficult to follow. In particular, I'm not sure that I could train a BPNN given only the content of the main paper. I understand that it is difficult to trim a lot of experiments into a short page limit, but what is left in the main text leaves way more questions that answers. This is true for some of the model discussion and applies two-fold for the experimental evaluation. - Could you include some details about how to actually train this model in the main text as this seems to be the most important step of the whole process. - I'm confused a bit in the experimental section with respect to what version of BP is actually being compared against. When you say "standard BP", you mean without damping? If so, I don't think this is a fair comparison as most practitioners would not use the undamped version. Additionally, why would we expect BPNN-D to converge to better local optima than BP? Because the learned H function helps avoid local optima? If this is true, it is an interesting observation by itself. - Plots and figures should probably include variance estimates of the RMSE to give some sense of the variability of the methods. - I liked the experiments in Section 4.3, but I think that it would have been more interesting to train on one of the domains and predict one of the others. In particular, I'm very curious how overfit (if at all) these models are to the domain on which they are trained. Maybe the corrections to the message passing procedure are really only functions of the graph structure and potentials...

Correctness: I wasn't able to verify all of the results, but it appears to be mostly correct.

Clarity: - "equivariant" should probably be properly defined. I found this section quite difficult to read compared to the earlier ones. - The variables don't need to be binary, correct? If so, then Thm 3. should apply to all log-supermodular models without the binary restriction. See: Nicholas Ruozzi. Beyond log-supermodularity: lower bounds and the Bethe partition function. Uncertainty in Artificial Intelligence (UAI), July 2013. - Outside of other clarity issues mentioned above their only a few minor typos.

Relation to Prior Work: The work seems to be appropriately positioned.

Reproducibility: No

Additional Feedback:


Review 2

Summary and Contributions: This paper generalizes Loopy Belief Propagation (LBP) on factor graphs by augmenting the architecture with learnable neural operators that are used to compute the aggregated factor-to-node messages. The paper proves some desirable properties that guarantee exact results on loop-free factor graphs. The effectiveness of the proposed technique is evaluated on 3 relevant settings.

Strengths: The theoretical grounding looks sound and the proposed technique shows promising results on different tasks (partition function estimation on Ising and Stochastic Block models, propositional model counting of formulas in CNF). The empirical evaluation is solid and shows remarkable performance gains with respect to state-of-the-art approaches. I also found he paper very well written.

Weaknesses: My only complaint is that there's no mention in the paper about releasing the code publicly.

Correctness: The empirical evaluation looks very solid, although I am not an expert in the area and I couldn't carefully verify the theoretical claims.

Clarity: Yes.

Relation to Prior Work: I think so, although I'm not familiar with this area of research.

Reproducibility: Yes

Additional Feedback: Minor: L.142-143 : "Variables are INDEXED (X 1 , X 2 , . . . , X n ) AND LOCALLY (by factors that contain them) in the representation of a factor graph." Do you mean "globally and locally"? --- I thank the authors for their feedback. I'm glad to read that the code will be released. I agree with other reviewers asking for experiments on computing the marginals, but I still think that this paper contains significant and interesting results that should be published. I am keeping the same score.


Review 3

Summary and Contributions: Proposes a generalization of the belief propagation (BP) algorithm that modifies the message-passing updates with a learned function (parameterized by a neural network), designed in such a way that the fixed points of BP are provably preserved, but learning allows BP to converge to better fixed points more rapidly. A further generalization changes the BP fixed points, but allows more flexibility to improve partition function estimates for particular model types. Experiments show accuracy improvements over conventional BP and standard graph neural networks (GNNs) for various factor graphs, primarily ones that arise in constraint satisfaction problems. POST-REBUTTAL: I agree with other reviewers that adding further details about BPNN training is important for reproducibility. After further consideration, I think the biggest limitation of this work is the lack of focus on marginal estimation. The rebuttal points out a relevant experiment, but because the training loss does not consider marginals, it is not clear (given existing theory) why we should expect BPNN to produce better marginals.

Strengths: The construction of a flexible family of tunably-damped BP message updates (BPNN-D), which are guaranteed to preserve BP fixed points, is elegant. Clear proofs of correctness are included in the supplement. This primary contribution is supplemented by neural network layers that learn to predict the partition function in a way that generalizes the Bethe free energy, and by message update layers that change BP fixed points for potentially greater accuracy (but with fewer guarantees). Experimental significance and insights: 1) For Ising models, there is good evidence of improvements in convergence frequency, and ability to sometimes find better fixed points than standard BP updates. 2) For stochastic block models, updates that change BP fixed points have notably higher accuracy than either BP or standard GNNs. A nice comparison suggests the improvement over GNNs is due to the way in which standard BP updates avoid "double-counting" messages, which BPNN preserves, but symmetric GNN updates do not. 3) For model counting problems where standard BP updates are known to suffer from convergence problems, BPNN-D updates give estimates that are more accurate than baseline algorithms with similar computational speed.

Weaknesses: While there are nice technical contributions here for a narrow set of models, a few factors may limit impact: * For the free energy prediction of Eq. (7), the "brute force" sum over the factorial number of permutations of the variables in each factor will limit application to low-degree models, while BP can be applied to some high-degree graphs whose factors have tractable structure. * Experiments focus exclusively on constraint satisfaction models, or models (like stochastic block models) which are defined by "soft" constraints. For such models, failure to converge is known to be a major issue for BP, and the BPNN-D dynamics have room for improvement. But for many other types of models, the larger issues are around accuracy of BP fixed points rather than convergence dynamics, and this approach may not be provide benefits. * To train the BPNN algorithm, a family of related models (about 50 instances) with true partition functions is required. This means the method is only useful when many closely-related problems need to be solved. The "generalization" demonstrated in the experiments in the paper is underwhelming (e.g., from 10x10 to 14x14 Ising models). * The training and experiments focus on the prediction of partition functions, which is indeed sometimes useful. But in practice, BP is more frequently used to compute approximate marginals of variables, and this is not addressed by the work here. For the variant of BPNN that modifies BP fixed points, there seem to be unexplored connections to other generalized BP algorithms that modify message passing updates to boost accuracy, like fractional/reweighted BP: * A new class of upper bounds on the log partition function, M. J. Wainwright, T. S. Jaakkola, and A. S. Willsky, IEEE Trans. Info. Theory, July, 2005. * Fractional Belief Propagation, Wim Wiegerinck, Tom Heskes, Neural Information Processing Systems 2002.

Correctness: Theorems in the main text are supported by proofs in the supplement, which seem correct. Some BPNN variants preserve more properties of BP than others, but the paper is reasonably clear about which is used in each case. I noted one small mistake: In Eq. (7), the inverse-factorial weight should be inside the sum over a. Evaluation metrics used in experiments are reasonable, and plots clearly illustrate results.

Clarity: While the subject matter is somewhat technical, and inaccessible to readers who don't know the variational view of loopy BP, overall the paper does a good job of explaining the contributions and their significance.

Relation to Prior Work: Citation of related work is reasonably good, but here is another paper that should be cited and discussed: Deep Unfolding: Model-Based Inspiration of Novel Deep Architectures John R. Hershey, Jonathan Le Roux, Felix Weninger https://arxiv.org/abs/1409.2574

Reproducibility: Yes

Additional Feedback:


Review 4

Summary and Contributions: The authors propose Belief Propagation Neural Networks (BPNNs) which can be considered to be a generalization of Pearl’s Belief Propagation (BP) algorithm as well as this procedure’s damped version. In particular, BPNNs are a special case of Graph Neural Networks (GNNs) with the modification, that they take factor graphs as inputs. BPNNs are composed of two parts: the first part are Iterative Layers which allow for nonlinear factor-to-variable message updates by introducing a learned operator H which replaces the conventional damping parameter and thus acts on differences between messages corresponding to successive iterations of BP; the second part is a so called Bethe free energy layer consisting of two Multilayer Perceptrons (MLPs). In the experiments, BPNNs are applied to three different types of models: Ising Models, Stochastic Block Models and counting of SAT solutions.

Strengths: As damping can often improve the convergence properties of BP, it is an interesting idea to introduce kind of an adaptive nonlinear operator instead of a manually chosen scalar damping parameter whose ‘best’ value is unknown in general. This provides a more flexible framework for message updates and therefore automatically solves this issue, how to select this parameter. We also want to remark positively that BPNNs are able to maintain some of BPs standard properties (e.g. providing the exact solution on tree structured factor graphs) and therefore maintain a theoretical connection to the original version. However, the theorems in the paper are essentially trivial and do not add anything in understanding when BP on loopy graphs converge.

Weaknesses: The absence of any explaining figures makes it hard for the reader to fully understand the underlying architecture of BPNNs. Specifically, it is not clear what the actual target of a BPNN acting on a particular factor graph is! Is it the partition function? Bethe Free energy? Or the beliefs? Moreover, the intention of the Bethe Free Energy Layer remains obscure. In lines 121-125 you mention: ‘When convergence to a fixed point is unnecessary, we can increase the flexibility [...] Additionally we define a Bethe free energy layer using two MLPs that take the trajectories of learned beliefs from each factor and variable as input and output scalars.’ In what situation is convergence to a fixed point unnecessary; convergence to fixed points is the actual goal of Belief Propagation and most of its modifications. On the other hand, the architecture of the Bethe Layer seems to be quite arbitrary. It seems so that BPNN (Eq 22) extends the classical BP equation by some nonlinearity using MLPs. This comes with the burden of additional hyperparameters for the NN and for training. Furthermore, an training phase is required. In this respect the comparision in the experiment is essentially unfair! In compares an approach what requires training with an approach which does not rely on any training! It compares apples with bears. Does the improved convergence time also reflect the training efford? Furthermore, there are many more advanced BP variants available which improve the convergence behavior at no additional costs (there was a workshop paper on one of the last NeurIPS Conferences – with the title “Self-guided Belief Propagation(?)” of similar) In the experiments, the models taken under consideration seem to be (at least partially) too restrictive. E.g. for Ising models only attractive pairwise potentials are admitted which makes inference in general much easier on the corresponding factor graphs, as frustrated cycles are absent.Furthermore, BP is limited to only 200 iterations. For the used large pairwise potentials, I assume that BP hardly converges at all. For large pairwise potentials it is known that BP does not work well. It would be good to see results for Cmax= {0.1, 0.5, 1}. Finally, for many practical applications (error correcting codes, medicine, ...) the estimation of the marginal distributions is much more important than an approximation of the partition function. In lines 825-840, however, the authors expose that ‘BPNN’s performance and generalization ability relative to GNN is not as strong as it was with estimating partitions, likely because it is not specifically trained to estimate marginals...’ which causes some serious doubts about the usefulness of the proposed architecture.

Correctness: Without going through all provided proofs in detail, the theoretical justification of the stated theorems seems to be correct. This includes in particular the proofs of the convergence properties of BPNNs on tree structured factor graphs and the necessary attention on isomorphisms between factor graphs.

Clarity: Several open questions remain for the reviewer after having read the current paper multiple times (details have already been discussed above). The specific BPNN model description is in some sense lacking – it is not really clear what the targets are. Furthermore, I found myself jumping back and forth between the appendix and the main text – hence the structure could be improved.

Relation to Prior Work: The relationship to other GNNs (GIN architecture) is discussed in the appendix.

Reproducibility: No

Additional Feedback:

[Author Response · NeurIPS 2020]

We thank all reviewers for their time and thoughtful comments. **Summary of positive reviews:** Reviewers liked our
proposed algorithm (BPNN) because it is innovative (@R1), elegant (@R3), interesting (@R4), and theoretically
grounded (@R2, @R4) with clear proofs (@R3). Our approach shows solid experimental improvements for computing
the partition function of Ising models, community detection problems, and factor graphs representing boolean formulae
(@R2, @R3, @R1). Additionally, we demonstrate robustness to perturbations in the test distribution (@R1). Reviewers
thought that our paper was generally well written (@R2) and that it does a good job of explaining our contributions
and their significance (@R3), although it is inaccessible to readers who aren't familiar with the variational view of
BP (@R3). **Summary of negative reviews:** The most serious concerns were that it was challenging to understand our
model's structure/output (@R4) and the training protocol we used (@R1). Additionally, there were questions/confusions
about our experimental setups.

**@R1, @R4: What is the output of BPNN?** BPNNs consist of two parts. First, iterative BPNN layers output messages,
analogous to standard BP. These messages are used to compute beliefs using the same equations as for BP. Second, the
beliefs are passed into a Bethe free energy layer (BPNN-B) that outputs an estimate of the log partition function. This
layer generalizes the Bethe approximation by performing regression from beliefs to $\ln(Z)$. Alternatively, when the
standard Bethe approximation is used in place of the (trainable) BPNN-B layer, BPNN provides additional guarantees.

**@R1: What is the training protocol?** In all our experiments, we initialized the BPNN to output the Bethe approxi-
mation obtained by running BP for a fixed number of iterations. We used the mean squared error between the BPNN
prediction and the ground truth log partition function as our training loss. When training BPNN-D without a BPNN-B
layer, we ran BPNN-D for a random number of iterations between 5 and 30 at every training step to encourage quick
convergence to a fixed point. (When training with a BPNN-B layer, iterative layers were applied for a constant number
of iterations, as convergence to a fixed point is no longer required.) We will highlight these details in the final copy.

**@R4: When is convergence to a fixed point unnecessary? How was the BPNN-B layer designed?** We can think
of BPNN as a trainable computation graph (neural network) that mimics the standard (unrolled) BP computation.
Convergence to a fixed point is unnecessary for good predictive accuracy; however, this extra flexibility makes theoretical
analysis more difficult. The Bethe layer (BPNN-B) was carefully designed to generalize the Bethe approximation while
maintaining invariance to factor graph isomorphism (please see Lemma 4 in the appendix).

**@R1, @R4: What version of BP is compared against? What does "standard BP" refer to?** We compared against
BP where we tuned the traditional damping coefficient and message update strategy (sequential or parallel) for best
results. "Standard BP" refers to belief propagation run with traditional damping but without learned modifications to
messages (via the operator $H(\cdot)$).

**@R4: Did you compare with a strong, baseline algorithm that utilized the same training dataset as BPNN?** Yes,
we compared with and found significant improvements over the Graph Isomorphism Network GNN.

**@R3, @R4: Did your experiments include problems where BP converges well?** Yes, BP converged on all commu-
nity detection problems and 88% of Ising models (line 178).

**@R1: "Why would we expect BPNN-D to converge to better local optima than BP? Because the learned H**
**function helps avoid local optima? If this is true, it is an interesting observation by itself."** Yes, that is the intuition
and we agree it is interesting! Fixed points of BP are local optima of the Bethe free energy. By training BPNN-D to
predict the exact partition function, BPNN-D can learn to find better local optimum.

**@R1: "I'm very curious how overfit (if at all) these models are to the domain on which they are trained."** We
explored cross domain generalization in our propositional model counting experiments and found that the quality of
results is dependent on the similarity of domains. We found that we could learn improvements that translate across a
broad variety of domains, although the improvements were less dramatic (lines 289-293).

**@R4: "The theorems in the paper ... do not add anything in understanding when BP on loopy graphs converge"**
That is correct, our theorems make no claims about the convergence properties of BP on loopy graphs. Our theorems
are given to *precisely characterize the relationship between fixed points of BP and BPNN-D* in terms of properties of
the learned operator $H(\cdot)$.

**@R4: "The theorems in the paper are essentially trivial ... Without going through all provided proofs in detail,**
**the theoretical justification of the stated theorems seems to be correct."** Even though the theoretical results may
seem intuitively correct in hindsight, they provide valuable insights into BPNN. We think it is unfair to characterize the
theorems as trivial because similar results do not hold for standard GNNs.

**@R3, @R4: Can BPNN be used to estimate marginals?** Yes. We focused on estimating the partition function
because these estimates are better understood for BP. However, in a small set of experiments on estimating marginals for
community detection, we found large improvements over BP and GNNs. This is a promising direction for future work.

**@R3: Can the BPNN-B layer be modified for computational efficiency on high dimensional factors?** Yes. The
BPNN-B layer can be modified to use an attention mechanism for efficient invariance given high dimensional factors.

**@R2: Will you open source your code after publication?** Yes!

[Meta-Review · NeurIPS 2020]

The reviewers felt the paper provides a valuable connection between modern graph neural networks and belief propagation. There was some confusion about how to train the BPNNs, and the paper should be revised to clarify these points. The authors should also expand on discussion about marginal estimation, even if is to highlight limitations of the proposed approach.